# Genome-wide analysis of dental caries and periodontitis combining clinical and self-reported data

Dmitry Shungin et al.[#]

Dental caries and periodontitis account for a vast burden of morbidity and healthcare spending, yet their genetic basis remains largely uncharacterized. Here, we identify self-reported dental disease proxies which have similar underlying genetic contributions to clinical disease measures and then combine these in a genome-wide association study meta-analysis, identifying 47 novel and conditionally-independent risk loci for dental caries. We show that the heritability of dental caries is enriched for conserved genomic regions and partially overlapping with a range of complex traits including smoking, education, personality traits and metabolic measures. Using cardio-metabolic traits as an example in Mendelian randomization analysis, we estimate causal relationships and provide evidence suggesting that the processes contributing to dental caries may have undesirable downstream effects on health.

---

The 2016 Global Burden of Diseases, Injuries and Risk Factors Study[1] estimated that dental caries in permanent teeth and periodontitis were the leading and 11th most prevalent causes of disease worldwide in 2016. Characterising the aetiology of these diseases is a priority as they confer major health[2] and financial burden, with the global cost of dental diseases exceeding 540 billion US dollars in 2015[3]. Part of this story is the genetic contribution to oral health outcomes and the heritability of dental caries and periodontitis have been reported to be as high as 50%[4,5]. However, the nature of this contribution remains poorly characterised despite the promise that increased understanding of genetic factors can bring with respect to aetiology dissection and theoretically, clinical management

Genome-wide association studies (GWAS) for dental caries have investigated measures, including overall caries experience[6,7], specific presentations of disease and the presence or absence of disease in paediatric populations[8,9]. To date, few reliable genetic-association signals have been found, and this is likely to reflect different measurement approaches used, complex genetic architecture[10] of dental caries or limited statistical power to detect associations. GWAS for periodontitis have investigated measures, including the presence or absence of disease[11], quantitative measures of periodontal status[12], severe presentations of disease[13], molecular and microbial intermediaries of disease[14] and composite phenotypes such as GWAS for principal components which aim to capture multiple facets of periodontal health[15]. These studies have not yielded consistent evidence of specific genetic contributions to periodontitis[9].

Currently, there is a trade-off between phenotypic refinement and obtaining valid phenotype data in sufficiently large cohorts to allow statistical power for genome-wide analyses. Given the logistical and economic challenges in obtaining detailed phenotypes in large populations, studies have focused on clinical outcomes indicating the presence or absence of manifest disease. Recent studies of other complex traits have taken a different approach and circumvented the limited availability of refined clinical phenotypes by taking advantage of extremely large, population-based, cohorts with both genetic data and less-refined proxy phenotypes. The UK Biobank (UKB), for example, includes information from approximately half a million participants[16]. The presence of less well defined but informative phenotypes has enabled successful association signal discoveries for several complex traits, yielding important biological insights[17]. In order to achieve further understanding of genetic contributions to caries and periodontitis, a shift in analysis scale—afforded by a similar approach to measurement—is required.

Here, we synthesise evidence from two sources; the Gene-Lifestyle Interactions in Dental Endpoints (GLIDE) consortium[18], which is a unique collection of epidemiological cohorts with detailed information on clinical endpoints of dental diseases; and UKB, which contains less-refined self-reported oral health data at a larger scale. In isolation, the biological meaning and relevance of the approximate measures in UKB would not be clear, but anchored against and unified with the more detailed clinical information in GLIDE, this combination of complementary resources provides a unique opportunity for genetic association discovery and application.

Using this combination of resources, we identify 47 novel risk loci for dental caries and show that the heritability of dental caries is partially shared with a range of complex traits. Applied analysis using Mendelian randomisation suggests that dental caries has undesirable downstream effects on health. Together, these findings improve our understanding of the potential causes and consequences of an important complex disease.

## Results

**Heritability and shared heritability of dental diseases.** Genome-wide association studies (GWAS) were undertaken within the GLIDE consortium, using clinical measures of dental diseases and imputed genotype data, accounting for age-specific, sex-specific and other study-specific covariates. The primary traits were measures of caries experience (the sum of Decayed, Missing and Filled tooth Surfaces (DMFS) and the sum of Decayed and Filled tooth Surfaces per available tooth Surface (DFSS)), number of teeth (Nteeth) and a dichotomous classification of periodontal health or disease (Table 1). All meta-analyses within GLIDE used a fixed effects inverse-variance-weighted strategy implemented in METAL[19] (Methods; Supplementary Data 1, Supplementary Data 2).

In parallel, genetic contributions to self-reported measures of oral health were characterised in UK Biobank (UKB). Participants were asked to report whether they had bleeding gums, painful gums, dentures, loose teeth, toothache or mouth ulcers as part of a baseline study questionnaire. GWAS for each trait was performed within individuals of European ancestry, accounting for age, sex and relatedness using a linear mixed model ($n = 461,031$ for all traits; see the Methods section)

For each dental disease trait, genome-wide single nucleotide polymorphism (SNP)-based heritability ($h2_{LDSR}$) was estimated using univariate linkage-disequilibrium score regression (LDSR)[20] (Methods). In GLIDE, the highest and lowest heritability estimates were reported for Nteeth (0.13, se = 0.02) and periodontitis (0.01, se = 0.01), respectively; while in UKB the highest and lowest heritability estimates were for dentures (0.09, se = 0.004) and toothache (0.04, se = 0.007), respectively (Fig. 1a).

To guide interpretation of dental disease traits and identify those with similar genetic contributions, genetic correlations ($R_g$) were estimated using bivariate LDSR[21] with an unconstrained intercept term to allow for sample overlap. For clinical measures within GLIDE, there were strong genetic correlations ($R_g > 0.4$ or $R_g < −0.4$) between DMFS and DFSS, DMFS and Nteeth and DFSS and Nteeth (Supplementary Table 1). Within UKB, each trait showed shared heritability with at least one other self-reported trait, but there was also evidence of distinct genetic architecture among the measures; for example, dentures had little genetic overlap with bleeding gums ($R_g = 0.0009$, se = 0.035) (Supplementary Table 2). Clinical and self-reported measures were then compared with identify pairs of traits with shared genetic contributions (Fig. 1b). For dental caries, the pair of clinical and self-reported traits with the greatest shared heritability were DMFS and dentures ($R_g = 0.82$, se = 0.087, $P = 4.1 \times 10^{-21}$ in Z-test). For periodontitis, the highest $R_g$ value was seen for loose teeth ($R_g \sim 1$); however, this was imprecisely estimated due to the low heritability of periodontitis (se = 0.78, $P = 0.17$).

**Novel risk loci discovered in meta-analysis.** Exploiting this shared heritability for gene discovery, single-variant association statistics from GLIDE and UKB were combined using a z-score genome-wide meta-analysis weighted by effective sample size. There were two principal analyses; one combining DMFS ($n = 26,792$ from nine studies) and dentures ($n_{cases} = 77,714$, $n_{controls} = 383,317$); the other combining periodontitis ($n_{cases} = 17,353$, $n_{controls} = 28,210$ from seven studies) and loose teeth ($n_{cases} = 18,979$, $n_{controls} = 442,052$). These analyses included ~8.9 million SNPs and insertion/deletions (INDELS) present in both GLIDE and UKB. There was evidence of a polygenic contribution to phenotype in both combined analyses with reasonable inflation given the nature of the traits and anticipated genetic architecture (genomic control factor [$\lambda_{GC}$] = 1.37,

**Table 1 Overview of dental disease phenotypes in GLIDE**

| Trait name | DMFS | DFSS | Nteeth | Periodontitis |
|---|---|---|---|---|
| Trait | Decayed, missing and filled tooth surfaces | Decayed and filled tooth surfaces standardised to number of tooth surfaces | Number of natural teeth | Presence or absence of periodontitis |
| Transformation | Standard deviation of tooth surface residuals | Inverse normal rank transformation | | — |
| Phenotypic assessment | Derived from clinical dental records | | Derived from clinical dental records or self-reported (WHGS) | Centers for Disease Control and Prevention/American Academy of Periodontology definitions[54] (four studies)<br>Two or more tooth surfaces with probing depth ≥5 mm, or at least four tooth surfaces with probing depth ≥4 mm (1 study)<br>Probing depth ≥5.5 mm in 2 or more sextants (1 study).<br>Participant-reported diagnosis of periodontitis (1 study) |
| Number of studies in primary analysis | 9 | 8 | 9 | 7 |
| Number of participants in primary analysis | 26,792 | 26,533 | 27,949 | 17,353 cases, 28,210 controls |

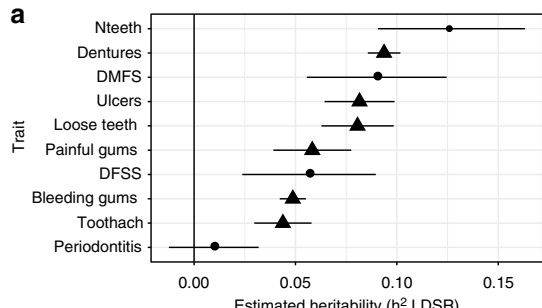

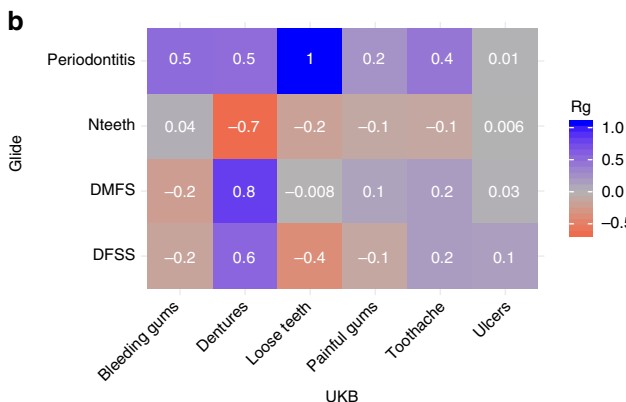

**Fig. 1** Estimated heritability of and genetic correlation between measures of dental disease. **a** estimated heritability ($h^2_{LDSR}$) for each trait in GLIDE (plotted in circles) and UKB (plotted in triangles). Error bars represent 95% confidence intervals. **b** Estimated genetic correlations ($R_g$) between traits in GLIDE and UKB. Cells are shaded according to the value of $R_g$

$h^2_{LDSR} = 0.085$, LDSR intercept $[LDSR_i] = 1.00$ for DMFS/dentures; $\lambda_{GC} = 1.09$, $h^2_{LDSR} = 0.046$, $LDSR_i = 1.00$ for periodontitis/loose teeth).

For DMFS/dentures, there were 47 conditionally independent genetic risk loci where at least one variant met a conventional

threshold for genome-wide significance ($P < 5 \times 10^{-8}$) in the unconditional results and following a stepwise selection procedure (Fig. 2a; Methods). The lead ten variants are provided in Table 2 and an extended table containing all variants is provided as Supplementary Data 3. The approximate conditional analysis procedure (Methods) also highlighted three risk loci containing multiple conditionally independent signals of association (Supplementary Table 3).

All 47 lead variants had directionally consistent effect estimates in GLIDE and UKB with good concordance in effect sizes (Pearson correlation coefficient = 0.94; Fig. 2b).

The largest estimated per-allele effect of any variant in the combined analysis was observed for rs121908120, a low-frequency variant (weighted Effect Allele Frequency [EAF] = 0.026 for the A allele, $P = 2.0 \times 10^{-22}$ in Z-test, standardised beta = −0.081, beta = −0.12 standard deviation (SD) change in DMFS residuals, odds ratio [OR] for having dentures = 0.85 (95% CI: 0.82, 0.88)). rs121908120 is a missense variant within *WNT10A* which results in a phenylalanine to isoleucine substitution and is predicted to have deleterious consequences in multiple *WNT10A* transcripts using the ExAC browser[22]. In a population of 28,691 adults aged 30–75 years who represent the Northern Swedish adult population with access to dental care, the estimated effect of the A allele corresponded to 2.1 (95% CI: 1.7, 2.6) fewer decayed, filled or missing tooth surfaces (Methods).

The variant with the strongest statistical evidence for association in the DMFS/dentures combined analysis was rs1122171, a common variant (weighted EAF = 0.59 for T allele, $P = 2.8 \times 10^{-62}$, beta$_{standardised}$ = 0.044, beta$_{DMFS}$ = 0.064, OR$_{dentures}$ = 1.09 (CI: 1.08, 1.10)), with an estimated effect corresponding to 1.2 (CI: 1.0, 1.4) additional decayed, missing or filled tooth surfaces per T allele. The lead variant lies within an uncharacterised protein-coding region, *C5orf66*, however, putative gene mapping using reference expression quantitative trait locus (eQTL) and chromatin interaction data implemented in FUMA[23] highlighted 25 potential genes at this locus (Supplementary Table 4).

The remaining 45 genetic risk loci in the DMFS/dentures combined analysis include some investigated in candidate gene

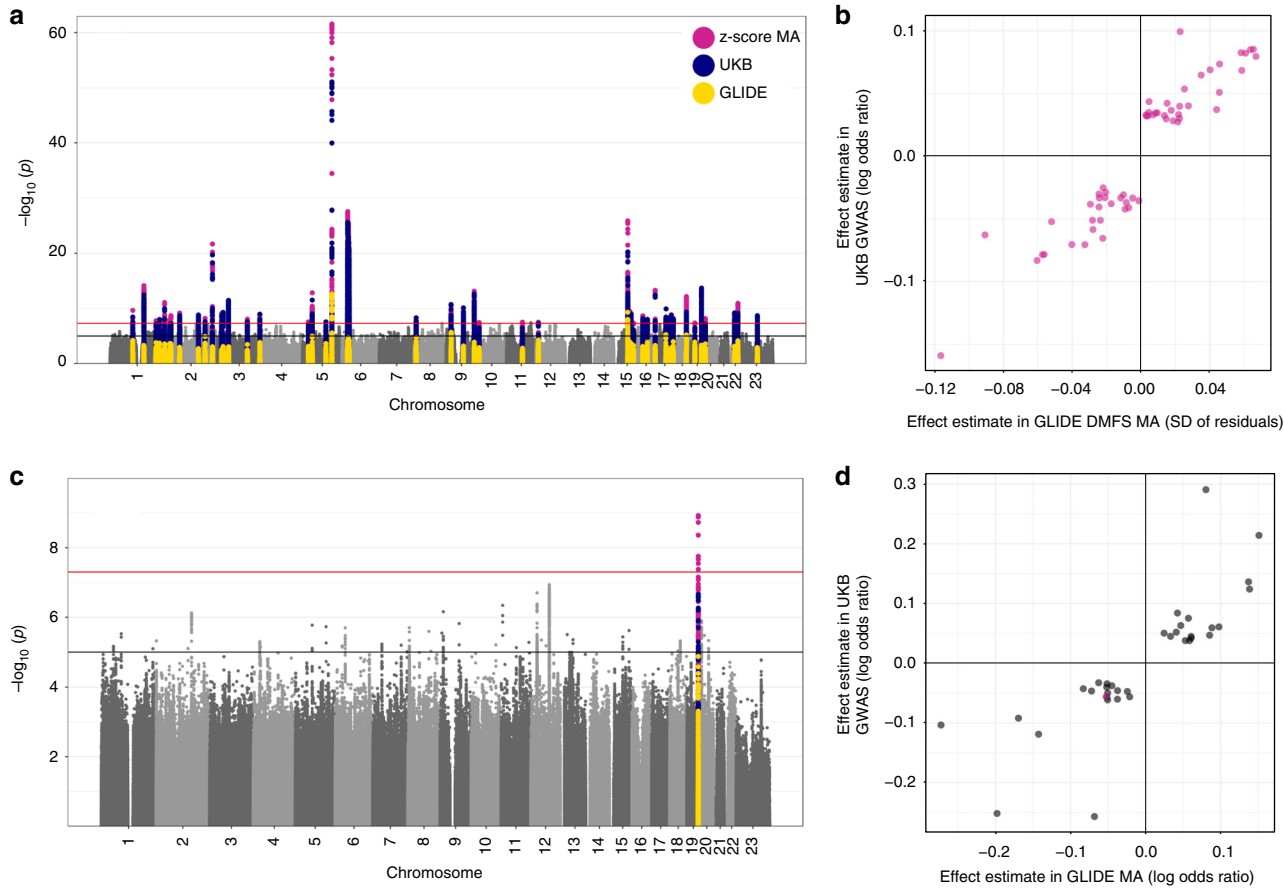

**Fig. 2** Single-variant results in combined analysis of GLIDE and UKB. **a** Manhattan plot of the DMFS/dentures combined analysis. The red line indicates the threshold for genome-wide significance at ($P = 5 \times 10^{-8}$), and the black line indicates a suggestive threshold for association at ($P = 1 \times 10^{-5}$). Loci achieving genome-wide significance in the combined analysis (Z-test) are coloured in magenta. $P$-values for the same loci are shown in blue for dentures in UKB and yellow for DMFS in GLIDE. **b** Concordance of genetic effects in the DMFS/dentures combined analysis. Each point represents a conditionally independent signal of association ($P < 5 \times 10^{-8}$). **c** Manhattan plot of the periodontitis/loose teeth combined analysis. The red line indicates the threshold for genome-wide significance at ($P = 5 \times 10^{-8}$), and the black line indicates a suggestive threshold for association at ($P = 1 \times 10^{-5}$). The single-locus meeting this threshold is coloured in magenta. $P$-values for the same locus are shown in yellow for periodontitis in GLIDE, and blue for loose teeth in UKB. **d** Concordance in genetic effects in the periodontitis/loose teeth combined analysis. The magenta point represents the single locus with $P < 5 \times 10^{-8}$, grey points represent conditionally independent suggestively associated loci with $P < 1 \times 10^{-5}$ in combined analysis

### Table 2 Ten lead novel independent loci in DMFS/dentures combined analysis

| Locus* | rsid | Chr:Pos (b37) | EA:EAF | DMFS/dentures | | DMFS | | Dentures | | N |
|---|---|---|---|---|---|---|---|---|---|---|
| | | | | Beta (se)† | P | Beta (se) | P | Beta (se) | P | |
| KRTCAP2 | rs4971099 | 1:155155608 | a:0.55 | −0.021 (0.0027) | 7.47E-15 | −0.024 (0.0089) | 0.006 | −0.040 (0.0055) | 3.24E-13 | 487821 |
| WNT10A | rs121908120 | 2:219755011 | a:0.03 | −0.081 (0.0083) | 2.03E-22 | −0.12 (0.039) | 0.0026 | −0.16 (0.017) | 1.94E-20 | 486339 |
| C5orf66 | rs1122171 | 5:134509987 | t:0.59 | 0.044 (0.0027) | 2.84E-62 | 0.064 (0.0087) | 2.40E-13 | 0.085 (0.0056) | 8.96E-52 | 487823 |
| FGF10 | rs1482698 | 5:44539453 | c:0.38 | 0.020 (0.0027) | 1.47E-13 | 0.023 (0.0092) | 0.014 | 0.040 (0.0057) | 3.10E-12 | 487823 |
| HLA | rs9366651 | 6:26336696 | t:0.51 | −0.029 (0.0026) | 2.66E-28 | −0.028 (0.0088) | 0.0014 | −0.058 (0.0055) | 4.47E-26 | 487822 |
| PBX3 | rs10987008 | 9:128661600 | a:0.64 | 0.021 (0.0028) | 7.47E-14 | 0.015 (0.009) | 0.093 | 0.042 (0.0058) | 2.52E-13 | 487822 |
| CA12 | rs72748935 | 15:63639416 | t:0.46 | −0.028 (0.0027) | 1.31E-26 | -0.052 (0.0092) | 1.60E-08 | −0.052 (0.0055) | 5.49E-21 | 487820 |
| FOXL1 | rs10048146 | 16:86710660 | a:0.81 | −0.026 (0.0034) | 5.20E-14 | −0.0236 (0.012) | 0.045 | −0.051 (0.007) | 3.85E-13 | 487820 |
| MC4R | rs28822480 | 18:57924823 | a:0.29 | 0.021 (0.0029) | 7.08E-13 | 0.044 (0.010) | 1.40E-05 | 0.037 (0.006) | 8.38E-10 | 487821 |
| MAMSTR | rs11672900 | 19:49220323 | a:0.47 | −0.020 (0.0027) | 4.67E-14 | −0.0092 (0.009) | 0.31 | −0.042 (0.0055) | 3.11E-14 | 487822 |

*The locus name refers to the protein-coding gene in the RefSeq database which is closest to the lead signal. †Estimates are from standardised regression coefficients

studies for dental caries, such as the Human Leucocyte Antigen (HLA) region[9], but none which have previously been identified directly in other GWAS for dental diseases. Using the PhenoScanner browser[24] (Methods) to make comparison with published studies, 46/47 variants or proxies ($r^2 > 0.8$, 1KGP phase 3, European ancestry reference data) were available in the PhenoScanner database, and 26 of these (57%) have previously reported association with one or more non-dental diseases or

health traits, termed 'overlaps' in this description. Traits seen repeatedly in this overlap analysis include adiposity traits (11/26 overlaps), height (9/26 overlaps) and bone mineral density (3/26 overlaps). A summary is shown in Supplementary Table 5, with full results for each variant in Supplementary Data 4. Regional association plots showing single-variant summary statistics and putatively mapped genes are provided in Supplementary Data 5.

In the periodontitis/loose teeth combined analysis there was evidence for association at a single-risk locus, *SIGLEC5*, which was recently reported as a risk locus for aggressive periodontitis[13] (Fig. 2c). The lead signal was seen at rs12461706, a common intronic variant within *SIGLEC5* (EAF = 0.40 for T allele, $P = 3.9 \times 10^{-9}$, $OR_{periodontitis} = 1.05$, $OR_{loose\ teeth} = 1.06$). Effect estimates in GLIDE and UKB were consistent at this lead locus, and there was also evidence for consistency in effect estimates at conditionally independent single variants passing an arbitrary threshold for suggestive association ($P < 1 \times 10^{-5}$; Fig. 2d).

**Association with predicted gene expression levels.** In an exploratory hypothesis-generating investigation, summary statistics from the two primary combined analyses and eQTL data were integrated using the S-PrediXcan[25] method to investigate the potential consequences of gene transcription on dental phenotypes. This method uses pre-trained models derived in data sets with measured gene expression to impute gene transcription. First, tissue-specific predictions were generated using models trained in GTEx[26] for 48 tissues, then the S-TissueXcan approach was used to integrate information across tissues into a single aggregate result for each gene transcript (Methods). In the DMFS/dentures, meta-analysis aggregate results were available for 15,522 transcripts. Of these, varying expression levels at 221 transcripts were predicted to influence the phenotype, including multiple HLA-region transcripts. Outside the HLA region, the strongest evidence was seen for *CA12*, located in the region of single-variant association signal at 15q22, ($P = 2.8 \times 10^{-17}$ in Z-test). *CA12* encodes a member of the carbonic anhydrase family of zinc enzymes which catalyse the hydration of carbon dioxide to form bicarbonate and hydrogen ions to regulate pH. This family has several functions relevant to dental caries, including tooth formation, where multiple carbonic anhydrases are produced by maturation-stage ameloblasts[27], salivary buffering, where defects in *CA12* lead to poor salivary function and xerostomia[28]; and regulation of tooth biofilm microbiota, where *CA6* may affect colonisation by the cariogenic microorganism *Streptococcus mutans*[29]. In the region of the lead single-variant signal in 5q31, the S-TissueXcan approach prioritised *PITX1* as the most relevant transcript ($P = 2.9 \times 10^{-12}$). *PITX1* encodes a developmentally expressed transcription factor with roles in skeletal and mandibular growth and tooth development, and deletion of the *Pitx1* locus in animal models results in abnormal mandibular tooth morphology[30]. The results are given in Supplementary Data 6. Full results for all transcripts are given in Supplementary Data 7 and shown in Supplementary Fig. 1.

In the periodontitis/loose teeth analysis, the same 15,222 transcripts were tested, but here only a single transcript was associated with the phenotype after correction for multiple testing. Increasing transcription of *SIGLEC5* was predicted to associate with increasing odds of periodontitis/loose teeth ($P = 8.7 \times 10^{-07}$). Full results are given in Supplementary Data 8 and shown in Supplementary Fig. 2.

**Tests for gene set enrichment.** Gene set enrichment was assessed using DEPICT[31], which tests whether loci housing dental disease-associated single variants are over-represented in predefined sets of functionally related genes. For the DMFS/dentures analysis, there was no evidence for enrichment in gene sets passing an acceptable false discovery rate (Supplementary Data 9), while for the periodontitis/loose teeth analysis there were insufficient independently associated loci to use this method. An alternative approach was implemented in FUMA[23] using a hypergeometric test to test whether the transcripts prioritised by S-TissueXcan were over-represented in curated gene sets defined in the Molecular Signatures Database (MSigDB C2). No enrichment beyond chance was observed after correction for multiple testing for DMFS/dentures loci, and there were insufficient loci to perform this analysis for periodontitis/loose teeth.

**Tests for enrichment in functional annotations.** Stratified LDSR was performed to test whether the heritability of dental disease traits was enriched for functional elements of the human genome[32] or in regions surrounding genes with tissue-specific expression patterns[33]. In the DMFS/dentures combined analysis, there was evidence for enrichment in 38/85 functional annotations including annotations providing evidence for evolutionary conservation. Treating Genomic Evolutionary Rate Profiling (GERP) scores as a continuous annotation, DMFS/dentures association signal was enriched in genomic regions under evolutionary constraint ($P = 1.5 \times 10^{-28}$ in Z-test for enrichment), while the 1.9% of variants annotated as highly conserved from primates to humans accounted for 30% of the heritability of DMFS/dentures (15.5-fold enrichment over baseline, $P = 3.0 \times 10^{-15}$; Supplementary Data 10). In LDSR analyses stratified by tissue annotation, the highest LDSR coefficients were seen for genomic regions specifically expressed in the gastrointestinal tract and minor salivary gland tissue; however, coefficients were imprecisely estimated, preventing tests for difference between tissue types (Supplementary Data 11). In the periodontitis/loose teeth analysis, there was little evidence for enrichment in functional or tissue annotation groups beyond chance (Supplementary Data 12, Supplementary Data 13), possibly reflecting limited statistical power for these analyses.

**Haplotypes in HLA region.** In the DMFS/dentures combined analysis, evidence for a complex pattern of association was seen at the HLA region of chromosome 6. To characterise this association signal, analysis of imputed HLA haplotypes was undertaken in UKB only ($n = 336,038$) using HLA haplotype dosage as the exposure and 'Dentures' as the outcome. Ten haplotypes were associated with dentures, including haplotypes of HLA class I and class II. The strongest evidence for association was observed for DQB1_201, a common haplotype encoding the DQ-beta 1 chain of the HLA class II complex (haplotype frequency = 0.15, OR = 1.07 (95% CI: 1.05, 1.09), $P = 8.9 \times 10^{-13}$ in $\chi^2$ test) (Supplementary Table 6). HLA class II molecules are expressed by antigen presenting cells, and alleles of HLA class II are thought to modulate the composition of the oral microbiome, including the cariogenic Gram-positive organism *Streptococcus mutans*[34].

**Population-specific sensitivity analyses.** The composition of the samples in GLIDE affords an opportunity to assess heterogeneity across strata of ancestry or clinical status. To explore whether lead variants identified the DMFS/dentures combined analysis had effects driven by periodontal tooth loss, genetic effect sizes were estimated in individuals with or without periodontitis and contrasted (GLIDE only). There was little evidence for different effect sizes between groups ($P_{FDR} >= 0.05$ in all tests for heterogeneity; Supplementary Table 7). Genetic effect estimates in the DMFS meta-analysis were compared between studies of Hispanic/Latino background (HCHS/SOL) and other European ancestry (GLIDE only) for lead associated single variants (Supplementary Table 8). There was little evidence for heterogeneity in effect sizes ($P_{FDR} >= 0.05$ in all tests for heterogeneity). For the periodontitis/loose teeth analysis, effect estimates at the lead variant were obtained from a meta-analysis of independent studies within GLIDE of East Asian ancestry ($N_{controls} = 15,670$, $N_{cases} = 1680$ from two studies). There was no strong evidence for association at rs12461706 in this population (OR = 1.03 for A allele, $P = 0.80$ in

$\chi^2$ test), where the A allele is common (EAF = 0.96) compared with the European ancestry population.

**Genetic correlations with other health-related outcomes**. Dental diseases are correlated with a range of adverse health traits and outcomes in observational studies, including acute endpoints of cardiovascular disease[35]. These associations were not only restricted to periodontitis but also seen for dental caries[36] and non-specific measures, such as tooth loss[37,38]. To evaluate whether these relationships were recapitulated at the genotype level, an analysis was undertaken using available external sources of information in the LD Hub resource[39] to screen for genetic correlations ($R_g$) between oral disease traits and all available traits with genome-wide association data, excluding any GWAS results obtained solely in UKB (Methods). For DMFS/dentures combined analysis, positive genetic correlations were seen for smoking traits, for example ever vs. never smoked ($R_g = 0.38$, se = 0.042, $P = 1.8 \times 10^{-19}$ in Z-test), adiposity traits, such as body mass index (BMI, $R_g = 0.21$, se = 0.027, $P = 3.2 \times 10^{-15}$) and smoking-related diseases, such as lung cancer ($R_g = 0.36$, se = 0.05, $P = 8.7 \times 10^{-13}$) and coronary artery disease ($R_g = 0.19$, se = 0.03, $P = 2.1 \times 10^{-10}$). Negative genetic correlations were seen for longevity (proxied by father's age at death, $R_g = -0.47$, se = 0.064, $P = 2.0 \times 10^{-13}$) and complex traits reflecting socio-demographics and cognition, in particular educational attainment ($R_g = -0.52$, se = 0.019, $P = 1.8 \times 10^{-163}$ for years of schooling) (Fig. 3; Supplementary Table 9, Supplementary Data 14). For the periodontitis/loose teeth combined analysis, genetic correlations were identified for 31 traits. Like DMFS/dentures, positive correlations were seen for cardiometabolic risk factors, including smoking and obesity, while negative correlations were seen for longevity and educational attainment traits (Fig. 4; Supplementary Table 10, Supplementary Data 15).

**Estimating potentially causal relationships**. The genetic correlations between dental diseases and metabolic traits and cardiac outcomes suggest shared genetic contributions to the same biological underpinnings, but also can be due to pathway effects where dental diseases are a risk factor for health outcomes. To explore this, we undertook bidirectional two-sample Mendelian randomisation (MR) to estimate causal relationships between DMFS/dentures and a subset of traits reflecting metabolic and cardiovascular health, using the Generalized Summary Mendelian Randomisation (GSMR)[40] approach for primary analysis. Treating DMFS/dentures as an exposure, a one standard deviation greater burden of dental disease (corresponding to ~20 additional decayed, missing or filled tooth surfaces) was nominally associated ($P < 0.05$) with greater waist-to-hip ratio (0.08 units after Inverse Normal Transformation (INT) (CI: 0.03, 0.13), higher plasma triglyceride levels (0.08 units INT (CI: 0.03, 0.13)), increased odds of type 2 diabetes (OR 1.2 (CI: 1.04, 1.38)) and increased odds of all stroke (OR 1.12 (CI: 1.01, 1.25)). Of these findings, only waist-to-hip ratio and triglycerides survived a Bonferroni correction for multiple testing. The primary results are given in Table 3.

Sensitivity analysis used alternative estimation tools to explore whether these findings were robust under different modelling assumptions (Methods, Supplementary Note 4). All effect estimates from IVW meta-analysis, MR-Egger regression and a heterogeneity-penalised model-averaging procedure were consistent with the primary estimates within 95% confidence intervals, apart from triglycerides, where the model-averaging procedure suggested a stronger causal effect of DMFS/dentures than indicated by the GSMR primary results (Supplementary Table 11).

Reciprocally, treating DMFS/dentures as an outcome, higher body mass index (BMI) and fasting blood glucose were nominally associated with greater burden of dental disease, but only body mass index survived correction for multiple testing (0.78 additional affected tooth surfaces (CI: 0.70, 0.86) for 1 $KgM^{-2}$ greater BMI; Supplementary Table 12). Estimates from sensitivity analyses were consistent with the estimates from the GSMR primary analysis within 95% confidence intervals for all traits apart from BMI, where the model-averaging procedure supported a stronger causal effect on DMFS/dentures than the GSMR primary estimate (Supplementary Table 13). Additional sensitivity analysis using multivariable MR suggested that BMI and fasting blood glucose had independent causal effects on dental disease (Supplementary Table 14).

Treating periodontitis/loose teeth as an outcome, higher BMI was associated with increased odds of disease (OR 1.05; 95% CI: 1.04, 1.06) for 1 $KgM^{-2}$ greater BMI, with consistent effect estimates in sensitivity analysis (Supplementary Tables 15, 16).

**Discussion**

This investigation used detailed clinical measures in combination with genetically validated proxy phenotypes to investigate the two major dental diseases, caries and periodontitis.

Combined analysis of DMFS and dentures identified 47 novel risk loci, including common variation near *PITX1*, *CA12* and in the *HLA* region in addition to uncommon variation with large effects in *WNT10A*. Combined analysis of periodontitis and loose teeth (a late symptom of advanced periodontitis) confirmed association at a previously reported locus, *SIGLEC5*. Integration of GWAS results with external sources of expression data prioritised 221 gene transcripts which may be involved in the pathogenesis of dental caries, while integration with functional annotation data showed that heritability of dental caries was enriched in conserved genomic regions, with a fold-enrichment value similar to other complex traits with serious health consequences[32]. Cross-trait comparisons showed overlap in the genetic determinants of dental diseases and complex non-oral health traits at both a single-variant level and genome-wide level. Finally, using a Mendelian randomisation analysis framework, there was evidence suggesting that biological processes leading to dental caries may have downstream effects on general health.

This investigation used an empirical approach informed by shared heritability to select dental disease proxy measures with similar biological meaning to detailed clinical measures. This approach has similarities to a recent investigation which used genetic correlation for post hoc confirmation of a prior expectation that family history of Alzheimer's disease is a satisfactory proxy of clinical diagnosis[41], but here we extend this approach to the selection of disease proxies where the prior expectation is less clear. This workflow may be adaptable to other complex diseases where it is difficult to obtain detailed phenotypic information in large collections.

At a single-variant level, the associated loci in this study may provide insight into the biological processes that are relevant to dental diseases. As an example, *WNT10A*, the locus harbouring the variant with the largest associated effect in the DMFS/dentures analysis, encodes a member of the WNT/β-catenin family of signalling proteins which have documented roles in inducing and regulating tooth formation in animal models[42–44] in addition to their widely known functions in embryological patterning and oncogenesis. In humans, *WNT10A* mutations have been reported to cause isolated defects in tooth number[45] and quality[46] and to regulate the cusp architecture and other morphological characteristics of teeth[47]. Genes encoding other proteins in the WNT signalling cascade (but not *WNT10A*) were identified as

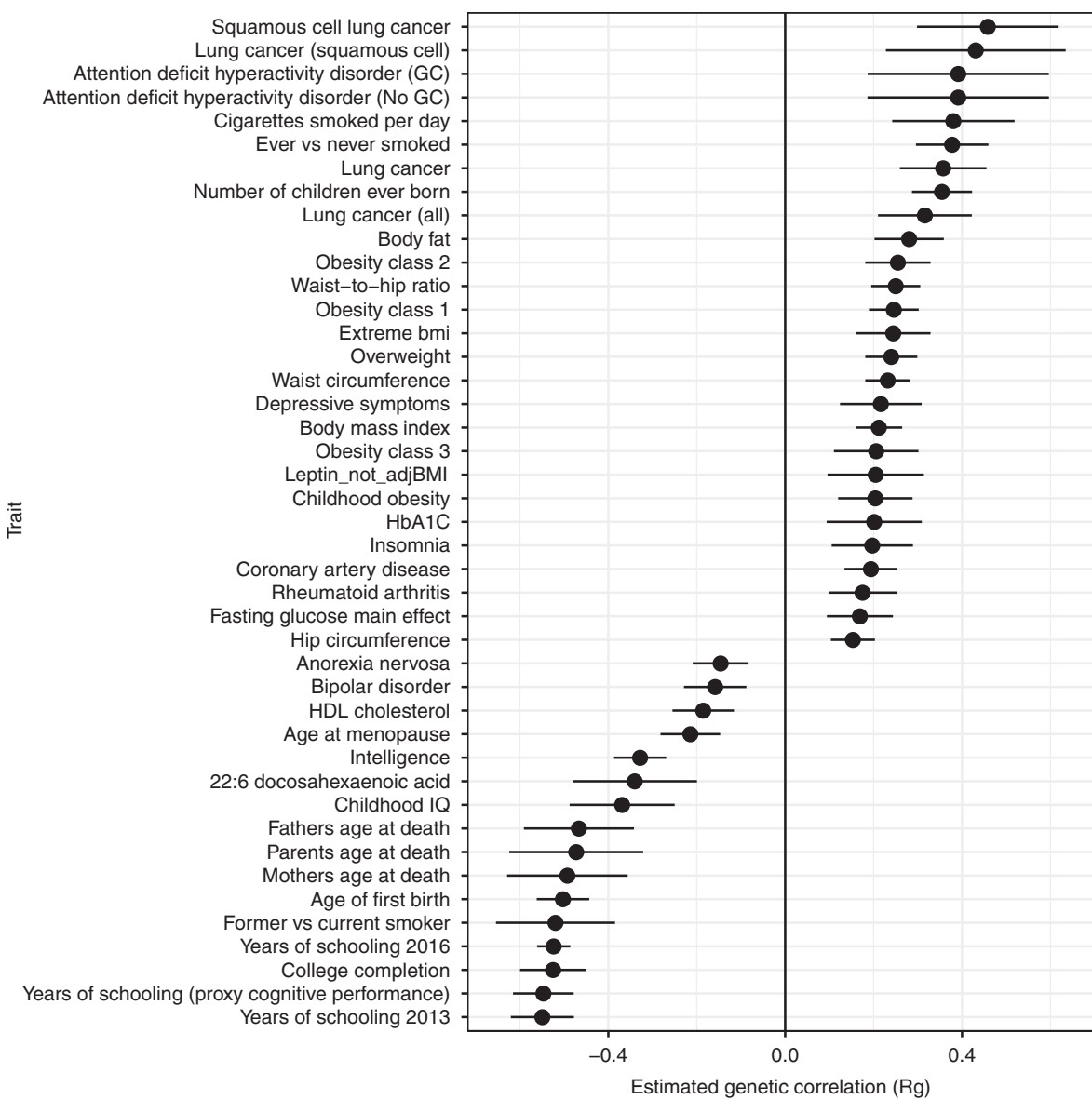

**Fig. 3** Estimated genetic correlations between DMFS/dentures and health traits or outcomes. Markers indicate the estimated magnitude of $R_g$, error bars represent 95% confidence intervals (1.96* LDSR standard errors on either side of the point estimate)

candidates in an early GWAS for dental caries with some sample overlap with this study[6]. Other examples with plausible biological relevance, including *HLA*, *CA12* and *PITX1*, are flagged in the results.

These genetic associations improve disease understanding and may provide a foundation for innovative approaches to risk assessment, outcome prediction, disease management or treatment. The availability of genetic proxies for dental diseases in large data sets also unlocks previously unavailable approaches, such as two-sample MR to infer potential consequences of dental diseases. As an illustrative example, the MR analysis included in this work provides some support for the hypothesis that dental diseases proxied by the DMFS/dentures may be an upstream risk factor for metabolic disturbance and cardiovascular disease events and prioritises specific relationships for additional investigation. This analysis also provides a broader demonstration of the utility of these data for generating and testing research hypotheses.

The dental traits examined in this study appeared moderately heritable, except for periodontitis where the quantifiable heritable signal was much lower than estimates reported in the

literature[5,48]. This might be due to heterogeneity introduced by the different approaches to disease classification, different patterns of periodontal treatment and varying distributions of age in the GLIDE cohorts or gene–environment interactions not accounted for in the study design. Here, DMFS/dentures is described as a measure of dental caries. This measure might capture some variation in periodontal status, as periodontitis is a major cause of tooth loss[49]. Conversely, we note that the *SIGLEC5* periodontitis risk locus is not a main finding for DMFS/dentures and that there is little evidence for heterogeneity in DMFS effect sizes between periodontal cases and controls, supporting the idea that the two primary meta-analyses capture distinct dental diseases.

The combination of genetically similar but non-identical pairs of phenotypes creates challenges in harmonising effect sizes, which we addressed using standardised regression coefficients, allowing valid comparison of the relative magnitude of genetic or causal effect estimates and valid inference about the probability of a non-zero effect. Indicative estimates on a tooth surface scale are also provided, however, these rely on assumptions about the

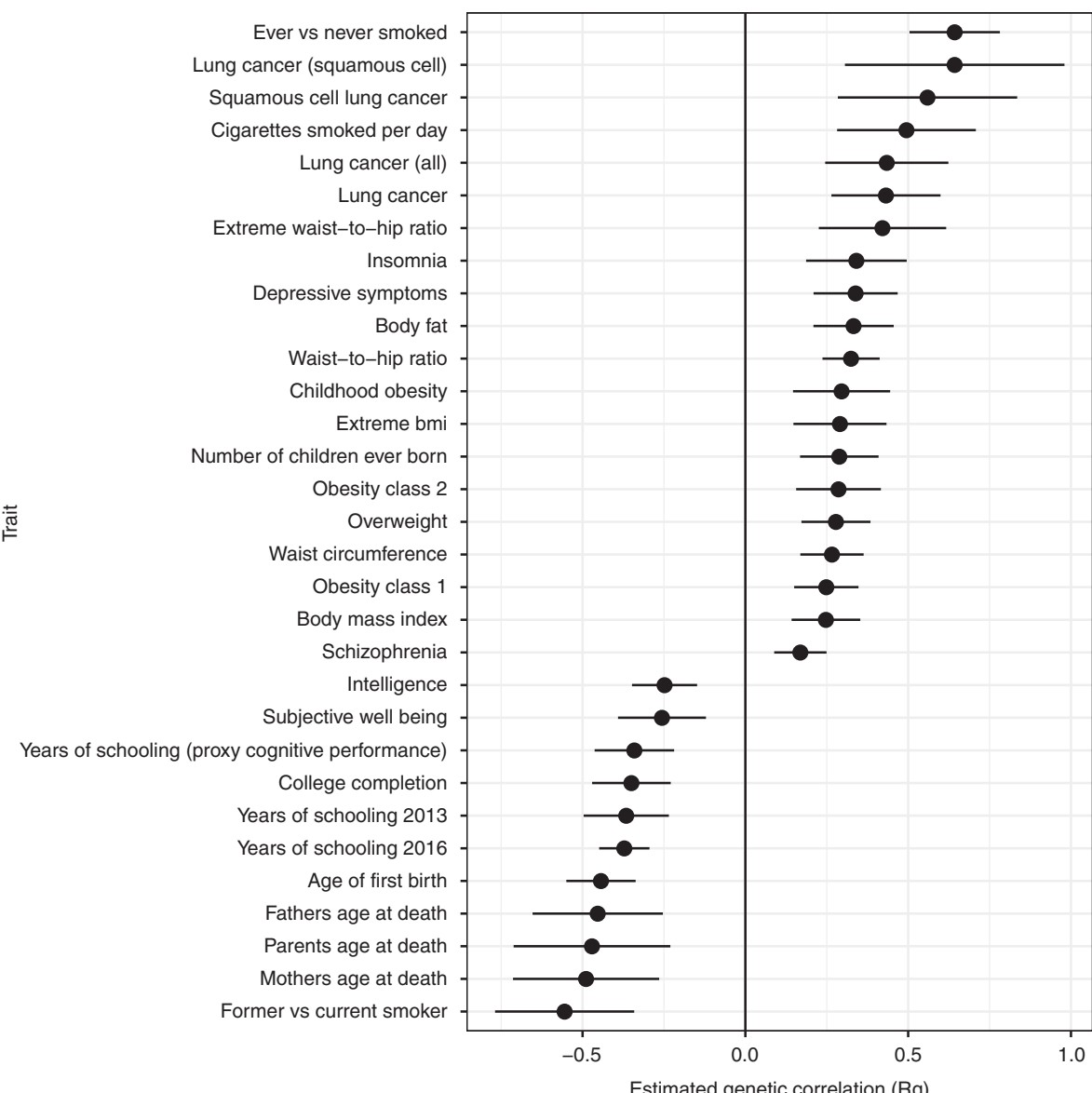

**Fig. 4** Estimated genetic correlations between periodontitis/loose teeth and health traits or outcomes. Markers indicate the estimated magnitude of $R_g$, error bars represent 95% confidence intervals (1.96* LDSR standard errors on either side of the point estimate)

| Table 3 Estimated causal effect of a one standard deviation greater lifetime burden of dental disease as measured by DMFS | | | | | |
|---|---|---|---|---|---|
| **Trait** | **Untransformed beta (se)** | **Transformed beta (95% CI)** | **Units for outcome** | **nSNP** | **P** |
| BMI | −0.006 (0.019) | −0.02 (−0.14, 0.10) | $KgM^{-2}$ | 35 | 0.75 |
| Waist-to-hip ratio adjusted for BMI | 0.11 (0.036) | 0.078 (0.027, 0.13) | SD INT | 45 | 0.0025 |
| Type 2 diabetes | 0.25 (0.10) | 1.20 (1.04, 1.38) | OR | 51 | 0.014 |
| Fasting glucose | 0.004 (0.034) | 0.0026 (−0.045, 0.051) | mM | 54 | 0.92 |
| HDL-c | −0.030 (0.041) | −0.021 (−0.078, 0.036) | SD INT | 44 | 0.47 |
| LDL-c | 0.023 (0.043) | 0.016 (−0.044, 0.076) | SD INT | 44 | 0.60 |
| Triglycerides | 0.11 (0.038) | 0.080 (0.027, 0.13) | SD INT | 45 | 0.0032 |
| Coronary artery disease | 0.13 (0.079) | 1.10 (0.98, 1.23) | OR | 54 | 0.092 |
| All stroke | 0.16 (0.077) | 1.12 (1.01, 1.25) | OR | 55 | 0.037 |
| *P*-values are obtained from the GSMR test | | | | | |

variance in DMFS scores in any given population and should be interpreted with caution.

GWAS results were explored in follow-up analyses using approaches which exploit external sources of information, including gene expression data. While these approaches maximise value from GWAS and improve statistical power to identify relevant genes[50], they are limited by the quality and availability of the external data, for example the data on odontoblast or ameloblast tissue are not available in GTEx. Cross-trait comparisons of dental diseases with other traits at single-variant level are only

possible when GWAS results are available for any given trait, and well-powered investigations are likely to report more associations than small studies. Together, this means that the list of observed overlaps between dental diseases and other traits will be biased towards well-investigated traits and outcomes.

Causal effect estimates in MR experiments may be biased by unaddressed horizontal pleiotropy[51] or confounded if both the exposure and outcome are influenced by a latent shared trait with genetic determinants[52]. The different approaches to detect and account for horizontal pleiotropy used in this study yielded similar causal effect estimates; however, lack of precision in these estimates may have masked important differences or evidence for horizontal pleiotropy which will only emerge in future experiments with greater statistical power. Finally, the non-specific-nature of the DMFS/dentures phenotype may capture a range of latent treats upstream of clinically manifest disease and tooth loss due to periodontitis, meaning that the downstream effects of DMFS/dentures might involve a wide range of potential mechanisms or mediators.

In summary, this study leveraged self-reported dental disease proxies and clinical disease measures in meta-analysis to identify novel risk loci for dental caries. Characterising the functional biology at these loci may provide insight to improve diagnosis or guide the development of new therapeutic approaches targeting the most prevalent global disease. At a single-variant and whole-genome level, the heritability of dental caries partially overlaps with other complex traits and diseases, arguing against the notion that dental diseases can be diagnosed, managed or researched in isolation. Finally, the results of MR analysis support the idea that population-level interventions to reduce the burden of dental diseases may have benefits for population health more generally.

## Methods

**Ethics statement**. Each contributing study obtained informed consent from all study participants. This study complied with all relevant ethical regulations, including the Declaration of Helsinki, and ethical approval for data collection and analysis was obtained by each study from local boards as described in Supplementary Data 1.

**Overview of study design and participants in GLIDE**. The contributing studies were (1) Atherosclerosis Risk in Communities (ARIC); (2) The Center for Oral Health in Appalachia cohort 1 (COHRA1), which is part of the GENEVA caries consortium; (3) the Dental Registry and DNA Repository of the University of Pittsburgh School of Dental Medicine, (DRDR), also part of the GENEVA caries consortium; (4) the Hispanic Community Health Study/Study of Latinos (HCHS/SOL); (5) the Malmö Diet and Cancer Study (MDC); 6) the Northern Finland Birth Cohort 1966 (NFBC 1966); (7) the Study of Health in Pomerania (SHIP), (8) the Study of Health in Pomerania Trend (SHIP Trend); (9) TWINGENE, which is a genotyped epidemiological study recruited from the Swedish Twin Registry (TWINGENE); (10) the Women's Genome Health Study (WGHS); (11) Biobank Japan (BBJ) and (12) Tokyo Medical and Dental University Aggressive Periodontitis Study (TMDUAGP). A detailed description of each study is included in (Supplementary Data 1 and 2).

In nine studies, analysis was conducted in individuals of European ancestry (ARIC, COHRA1, DRDR, MDC, NFBC1966, SHIP, SHIP-TREND, TWINGENE and WGHS). In one study (HCHS/SOL), participants were recruited from Hispanic and Latino communities in the USA, who self-reported ancestry from six broad groups (Cuban, Dominican, Mexican, Puerto Rican, Central American and South American). To undertake analyses within this highly admixed population, a bespoke modelling approach was undertaken. Multi-dimensional clustering was used to generate genetic analysis groups containing participants of similar ancestry. These group allocations were then used as covariates in a linear mixed model (partitioned to only fit the proportion of genetic structure due to familial relatedness rather than ancestry) alongside the first five genetic principal components, study center and log-transformed sampling weights[53]. Subsequently, the results from HCHS/SOL were treated as a study of European ancestry and included in the primary meta-analysis.

For periodontitis only, there were two studies with participants of East Asian ancestry (BBJ, TMDUAGP), totalling 17,287 participants. For periodontitis, separate meta-analyses were performed for studies of European ancestry and studies of East Asian ancestry.

**Dental disease traits in GLIDE**. Dental records used to calculate DMFS, DFSS and Nteeth were either obtained by a trained assessor as part of the study protocol, or from index linkage to records completed by a dental professional in primary care (MDC and TWINGENE). The examination protocols used by each study are described in Supplementary Data 1.

Nteeth was obtained by counting the number of remaining natural permanent teeth (excluding wisdom teeth, deciduous teeth, bridges or dentures). This was derived from records of clinical dental examination for all studies apart from WGHS, where participants were asked to self-report this information. Participants were asked whether they had lost any teeth and, if so, asked to specify the number of teeth lost.

For periodontal status, participants were classified as having (cases) or not having (reference) clinical symptoms of periodontitis, using definitions applied by each participating cohort. In ARIC, SHIP, SHIP-Trend and HCHS/SOL, criteria published by the Centers for Disease Control and Prevention/American Academy of Periodontology (CDC/AAP)[54] were used. In COHRA1, participants were classified as cases if two or more sextants had probing depth of at least 5.5 mm, or if participants reported ever having 'gum surgery'. In TwinGene and MDC, participants were classified as cases if at least two tooth surfaces had probing depth of 5 mm or deeper, or at least four tooth surfaces had probing depth of 4 mm or deeper. In BBJ, participants were classified as cases or controls based on clinical diagnosis by physicians at participating hospitals, with data retrieved from diagnosis codes in hospital registers. In TMDUAGP, the participants were classified as cases based on the classification developed at the 1999 international workshop for a classification of periodontal disease and conditions[55]. In WGHS, the participants reported if they had periodontal disease or not. The question stems were 'Since you started the trial (around 3 years ago)/In the past year, were you newly diagnosed with/have you had any of the following', with 'Periodontal disease' as one possible response. Participants who selected this response were asked to provide the month and year of diagnosis. Participants with no teeth were excluded from scoring the primary oral health traits in GLIDE.

**Genotyping in GLIDE**. Genotyping was performed using commercially available arrays (Supplementary Data 1). Prior to imputation, all genotypes were aligned to Build 37 SNP positions, and sample quality control (QC) measures were applied within each study as described in Supplementary Data 2. Imputation was performed using the 1000 Genomes reference panel[56] (1000 Genomes project (1kGp) phase 1 version 3 (March 2012) release for all studies apart from COHRA, which used the 1000 Genomes project phase 1 version 2 release).

**Single-variant tests for association in GLIDE**. For quantitative traits (DMFS, DFSS and Nteeth), derived variables were created to account for major covariates. Raw phenotype counts were regressed on age, age squared, genetic principal components and other study-specific covariates as described in Supplementary Data 2. Residuals from these regressions were transformed to z-scores. For DFSS and Nteeth (but not DMFS), residuals had a markedly non-normal distribution, so inverse normal rank transformation was applied. These stages were performed separately in male and female participants except for family-based studies, where sex was instead included as a covariate in phenotype preparation. For the binary trait of periodontitis, age, age-squared and other study-specific covariates were instead included as covariates in association tests. Every genotyped and imputed SNP was tested for association with these transformed variables using linear regression and additive genetic models, implemented via several software tools (Supplementary Data 2).

**Central QC measures**. In addition to study-specific measures, central QC of the association result files was performed using EasyQC software and a previously published protocol[57]. Variants were excluded if they had low imputation quality (below 0.3 for MACH, 0.4 for IMPUTE), departed from Hardy–Weinberg equilibrium ($P_{HWE} < 10^{-6}$ in the entire sample for continuous traits or in controls for periodontitis), or had per-file minor allele count <6. Directly genotyped SNPs with poor call rate (<95%) were also excluded. Association statistics within each study were corrected using genomic control inflation factor (max $\lambda_{GC} = 1.07$).

**Meta-analysis of studies in GLIDE**. Fixed effect inverse-variance weighted meta-analyses were performed using METAL software[19].

**Heterogeneity assessment in GLIDE**. For lead associated variants, DMFS effect sizes were compared between periodontitis cases and controls in stratified meta-analysis. Tests for difference in effect size were performed using the CALCPDIFF function of EasyStrata[58]. The HCHS/SOL study contains participants of Latino/Hispanic ancestry. To assess for ancestry-related heterogeneity, meta-analyses in GLIDE were repeated, excluding HCHS/SOL, and tests for heterogeneity were performed between these results and results from HCHS/SOL for the lead SNPs, again using EasyStrata.

**Analysis of chromosome X in GLIDE**. Where available, genotype data for chromosome X were analysed. Haploid allele calls for male participants were coded as 0

or 2. Genotypes for female participants were coded as 0, 1 or 2 under the additive genetic model. Summary statistics from the participating studies were combined using fixed effects, inverse-variance weighted meta-analyses following the same workflow described for autosomal results.

**Participants in UKB.** UK Biobank is a population-based health research resource consisting of ~500,000 people, aged between 40 and 69 years, who were recruited between the years 2006 and 2010 from densely populated regions of England, Scotland and Wales[16].

**Genotyping in UKB.** Participants in UKB were genotyped using the UK BiLEVE and UK Biobank axiom array (Supplementary Data 2). Pre-imputation quality control (QC), phasing and imputation were performed at the Wellcome Trust Centre for Human Genetics prior to data release[16]. Single-variant analysis used the 2018 (v3) imputed data release including imputation to both Haplotype Reference Consortium and UK10K/1000 genomes combined panel. Additional QC of the imputed data was performed at the University of Bristol. Participant-level QC involved restricting analysis to participants of European ancestry (identified as the largest cluster formed after k-means clustering on the first four genetic principal components, $N = 464,708$) as well as standard exclusions for mismatch between reported and genetic sex, possible sex chromosome aneuploidy, outliers in heterozygosity and missing rates. An additional two participants were removed as they were apparently related to a very large number of participants at 3rd degree or closer; otherwise related participants passing all other quality control were included in single-variant analysis[59]. For HLA haplotype analysis, imputation of classical HLA haplotypes was performed using the HLA*IMP:02 algorithm with a multi-population reference panel[16] and analysis was restricted to unrelated participants who self-identified as 'White British' and were located within the largest cluster in k-means analysis.

**Phenotypes in UKB.** At the baseline questionnaire, participants were asked 'Do you have any of the following? (You can select more than one answer)'. The possible answers included 'Dentures', 'Bleeding gums', 'Painful gums', 'Loose teeth', 'Toothache' or 'Ulcers'. Participants who selected one of these answers were coded as cases for each respective analysis. Participants who did not select this answer were coded as reference. GWAS included the following number of cases; dentures: $n = 77,714$ cases, bleeding gums: $n = 60,210$ cases, loose teeth: $n = 18,979$ cases, toothache: $n = 18,959$ cases, painful gums: $n = 13,311$ cases, ulcers: $n = 47,091$ cases.

**Tests for genetic association in UKB.** GWAS were performed using a linear mixed model (LMM) implemented in BOLT-LMM (v2.3)[60]. Age, age squared, sex and genotyping array were included as covariates in association testing. BOLT-LMM association statistics on a linear scale were transformed to log odds ratios and their corresponding 95% confidence intervals on the liability scale using a Taylor transformation expansion series as detailed in a full description of the GWAS pipeline which has been published online[61]. The association statistics were corrected using the estimated intercept term from univariate LDSR[20] (for dentures LDSR$_I$ = 1.0784, for loose teeth LDSR = 1.0432) prior to meta-analysis with GLIDE.

**Heritability estimation.** Univariate LDSR[20] was used to estimate heritability attributable to common genetic variants ($h^2_{LDSR}$). Scripts are available at (https://github.com/bulik/ldsc). Pre-computed reference LD scores and weights derived from 1000 Genomes European ancestry data were downloaded from (https://data.broadinstitute.org/alkesgroup/LDSCORE/). In GLIDE, summary statistics from the primary meta-analysis for each trait were used. A subset of high-confidence SNPs (MAF > 0.01, variants present in HapMap3) were used to minimise bias related to imputation accuracy. Summary statistics for each phenotype were first re-formatted with the munge_sumstats.py python script and then heritability was estimated using the–h2 option of ldsc.py.

**Genome-wide genetic correlations.** Genome-wide genetic correlations ($R_g$) for all pairs of traits between GLIDE and UKB were estimated using the–rg option of ldsc.py script (https://github.com/bulik/ldsc), and files were prepared using the same criteria described above. To estimate $R_g$ with a range of publicly available GWAS summary statistics, the prepared files for dental traits were uploaded to the LD hub[39] resource (ldsc.broadinstitute.org) using all available traits with genome-wide data available on 9th January 2019, apart from those obtained solely in UKB. The logical maximum and minimum values of $R_g$ are +1 and −1, however, the mathematical derivation used by LDSC can occasionally create values >+1 or <−1, which should be interpreted as values lying near +1 or −1, respectively.

**Partitioned heritability.** Heritability partitioned by functional annotation was estimated using the –ref-ld-chr argument of LDSR and pre-computed baseline and stratified models derived from 1000 Genomes phase 3 data (baselineLD_v2.1, November 2018 release). Segmented LDSR using tissue-specific annotations was

performed using the –h2-cts flag and multi-tissue gene expression files derived using 1000 Genome phase 3 data (April 2018 release).

**Combined analysis of UKB and GLIDE.** Combined two-way meta-analysis of association results between GLIDE and UKB was performed using the fixed effects z-score method implemented in METAL[19]. The contributions of GLIDE and UKB were weighted by sample size for continuous traits (DMFS) and by effective sample size for binary traits (dentures, periodontitis and loose teeth). The effective sample size was estimated from the number of cases and number of controls using the formula recommended by METAL's authors[19].

**Identification of lead single variants.** Genome-wide significance was set at $P < 5 \times 10^{-8}$. Within any given genomic locus, the lead SNP was defined as the variant with the lowest association $P$-value within a+/− 500-kb distance, using EasyStrata[58]. To test for conditional independence of lead variants, a stepwise selection procedure was performed using approximate conditional analysis[62], implemented using the cojo-slct function of GCTA (version 1.91.4)[63]. Loci were only considered functionally independent if at least one variant reaching genome-wide significance was selected in that risk locus during this procedure. For the HLA region (chr6: 25–35 Mb), the broad association signal and complex LD structure prevented testing for conditional independence between variants, so only a single lead variant was reported for this region. Where more than one conditionally independent variant was selected within a single-risk locus, the lead variant was presented in Table 1, with additional variants presented in Supplementary Material.

**Standardised regression coefficients.** To understand the relative size of genetic effects for variants in the Z-score combined analysis, beta coefficients and corresponding standard errors were derived using a transformation described in Supplementary Note 2.

**Indicative effect sizes.** Standardised regression coefficients in combined analysis were transformed to provide indicative effect estimates in a clinically interpretable scale of affected tooth surfaces or log odds of disease. The transformation is described in Supplementary Note 3.

**Tests for consistency of genetic effects of UKB and GLIDE.** Directional consistency of effects for lead in combined meta-analysis of GLIDE and UKB data was assessed by counting the number of directionally consistent SNPs between the GLIDE and UKB data sets. Correlation in effect sizes was assessed by calculating Pearson correlation coefficient weighted by inverse variance (1/SE) in the GLIDE meta-analysis. For periodontitis/loose teeth, there was only a single risk locus, therefore independent suggestively-associated single variants ($P < 1 \times 10^{-5}$) in the combined analysis were used to test for correlation in effect sizes.

**Cross-trait comparisons.** Cross-trait analysis of single variants was performed using PhenoScanner (http://www.phenoscanner.medschl.cam.ac.uk/phenoscanner), an online curated database of publicly available GWAS summary statistics results. The results were retrieved for variants identified in combined analyses and their proxies ($r^2 > 0.8$, 1 KGP phase 3, European ancestry reference data). If variants had previously been associated with a non-dental disease trait ($P < 5 \times 10^{-8}$ in GWAS), then details of the traits, association test statistics and publication references were retrieved.

**Gene set and pathway analysis.** DEPICT[31] (version 1.1, release 194), was used to test whether associated loci are preferentially located within predefined functionally similar gene sets, assessed against a null expectation informed by GWAS results for randomly distributed phenotypes. Associated loci were defined at ($P < 5 \times 10^{-8}$) in combined analysis. Reconstituted gene sets, tissue expression matrix and gene annotation files were downloaded from the DEPICT repository (data.broadinstitute.org/mgp/depict/documentation).

A hypergeometric test for gene set enrichment was performed using FUMA[23]. Associated genes were defined as those passing Bonferroni correction in S-TissueXcan analysis. Pathways and gene sets were taken from the Molecular Signatures Database (MSigDB C2), and all protein-coding genes in the FUMA database were used as background. Enrichment $P$-values were corrected for multiple testing using Bonferroni correction.

**Analysis of imputed gene transcription using S-PrediXcan.** Summary results from the combined meta-analyses were uploaded to the S-PrediXcan[25] web pipeline (https://cloud.hakyimlab.org/). Gene-level mediating effects of expression levels using were estimated using the summary GWAS association results and pre-fitted elastic net prediction models of gene expression levels in the 48 GTEx tissues[26], which are available online (http://predictdb.org/). Following this, multi-tissue results were aggregated using the S-TissueXcan[64] pipeline and post-processing pipeline (both hosted at (https://cloud.hakyimlab.org/). A Bonferroni correction was applied to account for the number of gene transcripts tested.

**Chromatin and eQTL mapping**. Associated single variants were mapped to candidate genes using the 'SNP2GENE' function in the online functional annotation tool FUMA[23]. eQTL mapping included data on cis-eQTLs (up to 1MB between SNP and gene) from all available tissues in GTEx v6 and v7. Chromatin interaction mapping included 21 tissues with HiC data and enhancer/promoter regions defined in Roadmap 111 epigenomes. By default, sites 250 bp upstream to 500 bp downstream from transcription factor start sites were considered promoter region windows.

**Tests for causal association**. Primary analysis was performed using Generalized Summary Mendelian Randomisation[40] (GSMR), implemented in GCTA (v1.92.0). Compared with other summary-statistic based estimators, GSMR is reported to have greater statistical power because GSMR uses genome-wide data to account for sampling variance in SNP-exposure and SNP-outcome estimates[40].

Genome-wide summary statistics for seven metabolic traits and two cardiovascular outcomes were selected as follows: BMI; meta-analysis of GIANT with UK Biobank[65]; for waist-to-hip ratio adjusted for BMI; GIANT consortium[66], for coronary artery disease CARDIOGRAM plus C4D meta-analysis[67]; for stroke the MEGASTROKE consortium[68]; for type 2 diabetes, the DIAGRAM consortium[69]; for fasting glucose the ENGAGE consortium[70]; for HDL cholesterol, LDL cholesterol and triglycerides, the GLGC data sets were used[71].

To achieve adequate variance explained and therefore statistical power, multiple SNPs were included as instrumental variables for all traits. All variants associated with dental disease or metabolic traits ($P < 5 \times 10^{-8}$) after LD clumping using reference data from the cohorts arm of the UK10K project (https://www.uk10k.org/data.html) to produce index SNPs ($r^2$ threshold, 0.01) were included as potential instrumental variables. No attempt was made to manually screen variants for possible undesirable pleiotropic effects, with filtering instead performed as part of the HEIDI-outlier procedure.

HEIDI-outlier filtering, is an extension of the heterogeneity in dependent instruments method[72]. This analysis projects a plausible distribution of causal effect ($\beta xy$) estimated from a non-outlying genetic instrumental variable, and tests whether other single nucleotide polymorphisms (SNPs) have values of $\beta xy$ compatible with this estimate, on the assumption that pleiotropic variants will have outlying values[40]. Following default criteria, potentially outlying variants with ($P < 0.01$ for pleiotropy) were removed and $\beta xy$ was estimated from the remaining instruments.

In the primary analysis, variants in the HLA region (chr6: 25–35 Mb) were removed for estimation of casual effects of DMFS/dentures, except for a single-lead variant. For estimation of causal effects of other traits on DMFS/dentures, full genome-wide data were used.

Methods used for sensitivity analysis are described in Supplementary Note 4.

**Reporting summary**. Further information on research design is available in the Nature Research Reporting Summary linked to this article.

## Data availability
Genome-wide summary statistics of the analyses have been deposited in an online repository and are available at: https://data.bris.ac.uk/data/dataset/2j2rqgzedxlq02oqbb4vmycnc2. Genome-wide summary statistics for all metabolic and cardiovascular outcomes used in this analysis were downloaded from the sources listed in Supplementary Note 1. Genome-wide summary statistics are available for the COHRA and DRDR projects' dental caries GWAS through the Human Genomics Analysis Interface of the FaceBase consortium (http://FaceBase.sdmgenetics.pitt.edu/, NIH Grant # 5U01-DE024425). Participant-level genomic and phenotypic data for the COHRA and DRDR projects are available through dbGaP (https://www.ncbi.nlm.nih.gov/gap/; dbGaP Study Accession #: phs000095.v3.p1). Access to UKBiobank data is through a manged open access procedure which is described in full online (http://www.ukbiobank.ac.uk/using-the-resource/). The data underlying Fig. 1a are provided in Supplementary Table 17, and the data underlying Fig. 1b are provided in Supplementary Table 18. Source data for Figs 3 and 4 are provided in Supplementary Tables 9 and 10.

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

## Acknowledgements

We are very grateful to all study participants and staff for their contributions to this research. Studies in GLIDE and the authors of this work receive funding from numerous sources who are acknowledged in full in Supplementary Note 1. This research has been conducted using the UK Biobank Resource (Application; 40644). The LDHub and PhenoScanner resources are made possible by studies and databases which made GWAS summary data available. These are listed in full online at http://ldsc.broadinstitute.org/about/ and http://www.phenoscanner.medschl.cam.ac.uk/information.html. We are grateful to all GWAS consortia for making data available on metabolic traits and cardiovascular outcomes, and these are listed in Supplementary Note 1.

## Author contributions

I.J., N.T. and P.W.F. conceived the study. D.I.C., P.M.R., Y.H.Y., P.M., T.S., Y.O., T.K., V. A., M.-L.L., M.O.-M., J.R.S., U.V., M.M., M.K., T.S., A.V., U.V., Y.F., S.O., K.E.N., C.A., P. W.F., N.T., E.I. and I.J. designed and supervised analysis. D.S., S.H., K.D., Y.K., M.K.L., K. G., G.H., V.A., P.P., A.T., B.T., S.S., J.H., F.G. and J.R.S. analysed the data. D.S., S.H., E.I., P.W.F., N.T. and I.J. drafted the paper. All authors reviewed, revised and approved the final version of the paper.

## Additional information

**Competing interests:** The authors declare no competing interests.

Dmitry Shungin[1,2,36], Simon Haworth[3,4,36], Kimon Divaris[5,6], Cary S. Agler[7], Yoichiro Kamatani[8], Myoung Keun Lee[9], Kelsey Grinde[10], George Hindy[11], Viivi Alaraudanjoki[12], Paula Pesonen[13], Alexander Teumer[14], Birte Holtfreter[15], Saori Sakaue[16], Jun Hirata[16], Yau-Hua Yu[17,18], Paul M. Ridker[17,19], Franco Giulianini[17], Daniel I. Chasman[17,19], Patrik K.E. Magnusson[20], Takeaki Sudo[21], Yukinori Okada[16], Uwe Völker[22], Thomas Kocher[15], Vuokko Anttonen[12,23], Marja-Liisa Laitala[12], Marju Orho-Melander[11], Tamar Sofer[19,24], John R. Shaffer[9,25,26], Alexandre Vieira[26], Mary L. Marazita[9,25,26], Michiaki Kubo[8], Yasushi Furuichi[27], Kari E. North[28], Steve Offenbacher[29,37], Erik Ingelsson[30,31,32], Paul W. Franks[33,34,35], Nicholas J. Timpson[3] & Ingegerd Johansson[1]

[1]Department of Odontology, Umeå University, Umeå SE-901 85, Sweden. [2]Broad Institute of MIT and Harvard, Cambridge, MA 02142, USA. [3]Medical Research Council Integrative Epidemiology Unit, Bristol Medical School, Bristol BS8 2BN, UK. [4]Bristol Dental School, Bristol BS1 2LY, UK. [5]Department of Pediatric Dentistry, School of Dentistry, University of North Carolina, Chapel Hill, NC 27599, USA. [6]Department of Epidemiology, Gillings School of Global Public Health, University of North Carolina, Chapel Hill, NC 27599, USA. [7]Department of Oral and Craniofacial Health Sciences, University of North Carolina, Chapel Hill, NC 27599, USA. [8]RIKEN Center for Integrative Medical Sciences, Yokohama, Kanagawa 230-0045, Japan. [9]Center for Craniofacial and Dental Genetics, School of Dental Medicine, University of Pittsburgh, Pittsburgh, PA 15219, USA. [10]Department of Biostatistics, University of Washington, Seattle, WA 98195, USA. [11]Lund University, Lund SE-223 62, Sweden. [12]Research Unit of Oral Health Sciences University of Oulu, Oulu FI-90014, Finland. [13]Infrastructure for Population Studies, Faculty of Medicine, University of Oulu, Oulu FI-90014, Finland. [14]Institute for Community Medicine, University Medicine Greifswald, Greifswald 17475, Germany. [15]Department of Restorative Dentistry, Periodontology, Endodontology, and Preventive and Pediatric Dentistry University Medicine Greifswald, Greifswald 17475, Germany. [16]Department of Statistical Genetics, Osaka University Graduate School of Medicine, Suita, Osaka 565-0871, Japan. [17]Division of Preventive Medicine, Brigham and Women's Hospital, Boston, MA 02215, USA. [18]Department of Periodontology, Tufts University School of Dental Medicine, Boston, MA 02111, USA. [19]Harvard Medical School, Boston, MA 02115, USA. [20]Department of Medical Epidemiology and Biostatistics, Karolinska Instituet, Stockholm SE-171 77, Sweden. [21]Department of Periodontology, Graduate School of Medical and Dental Science of Tokyo Medical and Dental University, Tokyo 113-8510, Japan. [22]Department of Functional Genomics, Interfaculty Institute for Genetics and Functional Genomics, University Medicine Greifswald, Greifswald 17475, Germany. [23]MRC, Oulu University Hospital and University of Oulu, Oulu FI-90014, Finland. [24]Department of Sleep Medicine, Brigham and Women's Hospital, Boston, MA 02130, USA. [25]Department of Human Genetics, University of Pittsburgh, Pittsburgh, PA 15261, USA. [26]Department of Oral Biology, University of Pittsburgh, Pittsburgh, PA 15213, USA. [27]Department of Oral Rehabilitation, Division of Periodontology and Endodontology, School of Dentistry, Health Sciences University of Hokkaido, Tobetsu, Hokkaido 061-0293, Japan. [28]Department of Epidemiology, University of North Carolina, Chapel Hill NC 27516, USA. [29]Department of Periodontology, University of North Carolina, Chapel Hill, NC 27599, USA. [30]Department of Medicine, Stanford University School of Medicine, Stanford, CA 94305, USA. [31]Stanford Cardiovascular Institute, Stanford University, Stanford, CA 94305, USA. [32]Stanford Diabetes Research Center, Stanford University, Stanford, CA 94305, USA. [33]Department of Clinical Sciences, Genetic and Molecular Epidemiology Unit, Lund University, Malmö SE-214 28, Sweden. [34]Department of Public Health and Clinical Medicine, Umeå University, Umeå SE-901 87, Sweden. [35]Department of Nutrition, Harvard T. H. Chan School of Public Health, Boston, MA 02115, USA. [36]These authors contributed equally: Dmitry Shungin, Simon Haworth. [37]Deceased: Steve Offenbacher.

