## [Peer Review File · Nature Communications]

Reviewers' Comments:

Reviewer #1:

Remarks to the Author:

Review of "Genome-wide analysis of dental caries and periodontal disease combining clinical and self-reported data"

This paper describes the largest study to date aiming to identify genetic determinants of Dental caries and periodontal disease. Whilst the sample size is impressive and many of the findings highly significant, I have a number of concerns with the manuscript:

1) The results section is very poorly written and structured, with many aspects of methodology unclear. For example, the number of genetic determinants is initially fixed at 17, but then changes or is re-assessed at half a dozen other occurrences throughout the results section. This leads to great confusion as to how many loci the authors are claiming and how robust the signals are. This inevitably leaves the reader navigating the huge number of poorly described supplementary tables in search of answers. I appreciate the complexity when trying to combine multiple traits, ancestries and analytical techniques, but the current approach is far from clear. I'd advise the authors consider what they feel is the most appropriate set of parameters and then present a single discovery analysis for the two traits. Appropriate sensitivity analyses can then be performed but always on the same number of X variants. It seemed slightly illogical to present MTAG as secondary/sensitivity when it identified so many more variants than the primary analysis. The authors should come to a consensus view on what they feel is robust or not without just leaving it to reviewers to navigate the many supplementary tables.

2) The number of defined independent signals in Table 1 looks concerning – presumably the multiple HLA signals are shaded in grey because the authors don't believe they're all truly independent? Perhaps the safest approach is to only report the most significant HLA association if this region can't be fine-mapped with more confidence. Can the authors also confirm that the remaining signals in that table are all uncorrelated with each other? (LD can range beyond 500kb). Was the associated HLA haplotype driving the association with any of the reported index variants or just coincidentally correlated?

3) The LDSC intercept of 0.81 for the DMFS/dentures discovery analysis suggests quite substantial over-correction of test statistics. On looking through the methods it appears that the UKBB results were corrected for a lambda GC of 1.46. This makes no sense at all considering the results have been generated from a linear mixed model and represents a severe over-correction. A more moderate approach would be to correct for the intercept value. Can the authors confirm what ancestry groups were included in the UKBB analysis and whether this included related individuals?

4) What version of the UKBB imputation was used? If not v3 (which I'm guessing due to X-chromosome omission), this should be rerun as it's likely to boost signal identification considerably.

5) Integration with eQTL data has been performed using FUMA and S-PrediXcan, however to my knowledge neither are evaluating co-localisation of signal. For example, are many of the overlaps with expression driven by coincidental overlap of LD rather than truly shared effects? It would also be informative to show how many transcripts map to each of the individually associated regions.

6) Many supplementary tables contain variant names which are cryptic and uninformative – this should be clarified.

7) Biological insight seems somewhat limited. The only pathway approach run is DEPICT, which didn't identify any pathways. Have the authors considered using other approaches?

8) What cell and tissue types were significantly enriched (most readily tested using LDSC-SEG)? This may be informative for downstream eQTL testing.

9) "Single-variant cross-trait lookup" – information in this paragraph very limited and points to verbose supplementary tables. Key information is what % of identified signals overlaps a non-oral health related complex trait.

10) Genetic correlations without follow up Mendelian Randomization are of limited value and likely mirror what you might observe in an epidemiological observational setting. At a minimum it would be informative to test the educational attainment and smoking bi-directional MRs in this manuscript.

11) Table 2 contains 3 data points and may be better served integrating into Table 1.

Reviewer #3:

Remarks to the Author:

In the paper under review, the authors perform a large-scale GWAS/meta-analysis on two dental traits, caries and periodontitis. The sample used for this analysis was composed of a well-characterized consortium project GLIDE, which provided extensive clinical data, and a more population-based and self-reported data set from the UK Biobank. With their study the authors suggest a possible approach to answer a very important question in complex genetics research: How can those large sample sizes be achieved that are required for the detection of low effect sizes? They test the combination of diverse measures as proxy for one specific trait, identify those pairs that provide the best genetic correlation, and maximize the sample size using these two measures from both cohorts. The approach worked very well for caries, while results are less clear / significant for periodontitis (see my comments). The authors well controlled for any potential biases when combining both data sets, the results (at least for dental caries) seem to be conclusive and reliable. However, it would be very interesting to the reader to see whether the approach of combining clinical samples and large-scale (less well characterized) samples has been previously performed by other groups / for other traits?

I have identified some critical points that mainly relate to some details of the analyses and restructuring of the results section. As major benefit, the paper under review generates the possibility for a lot of follow-up hypotheses in order to understand the biology underlying dental traits.

Major comments:

1) Analysis I: For the DMFS/DFSS analyses the authors included about 25.000 samples from several studies of which one (one of the smaller ones) was Non-European. This inclusion complicated the analyses (as two meta-analyses were done – one for Europeans, and one for “mixed ethnicities” with the contribution of the one additional sample). Similarly, two ethnicity-analyses were done for periodontal disease. Given the results I am not sure that this inclusion made sense – from about 489,000 individuals included in the DMFS/dentures analysis, only about 20,000 were from Hispanics.

As a consequence of this, the reader gets confused at several points as to which analysis the authors talk about. For instance, in the LDSR analyses (third part of the Results section), I assume it is European samples only? Or does this refer to the combined samples? As another example, page 11 (attenuation of the effect of rs12461706), it just becomes clear at the end of this page that all other analyses before were performed without the East Asians. I think that many of this information is provided in the Methods, but it would be very beneficial to have this included in the main text also (or, alternatively, to leave out the “mixed ethnicity” analyses).

2) Analysis II: As seen in Figure 1, the genetic correlation between loose teeth (UKB) and periodontitis (GLIDE) is 1. However, the heritability estimates provided in Figure 1a show significant differences between these two traits. This leaves me questioning whether there is a statistical error in some of these analyses, or what the explanation for this discrepancy is. Given these differences in heritability estimates, is the combination of these two traits in a meta-analysis justified?

3) Results/conditional analyses: The authors defined independent effects as variants that, after conditioning on the sentinel SNP, reached genome-wide significance, and which are located outside

a 500 kb region to either side. This approach is of high risk for false-negatives, as independent effects might be present within the 1 Mb around the lead SNP. What is the rationale for the 500 kb window? Also, it is questionable whether independent effects have to require genome-wide significance, given the limited power as compared to the first analysis – a decrease in P-values might be a good indicator, too. The authors should consider to re-run the conditional analyses and consider other organizational elements of the genome such as TADs for definition of possible boundaries.

Minor comments:

Abstract: The abstract would benefit from more details on the results – for instance, some highlights from the genetic correlation analysis or some of the interesting loci from the caries association study. At the moment the Abstract is more a project description and not very attractive to the reader.

Introduction: There is a number of superficial statements in the Introduction. Some examples: “Caries is the most prevalent disease worldwide” – this is hard to believe, and I was not able to find proof for this statement in the provided reference. How frequent is it (is it truly more prevalent than, for instance, cancer)? What is the evidence for this statement? As another example, the authors state that evidence for a heritable contribution exist “along with an obvious environmental burden”. What does “obvious” relate to? How much is the heritable estimates? Where do the numbers come from, and what is the nature of the environmental factors? Finally, referring to previous studies, the authors stated that they had “yielded interesting results”. What is known about the traits? Any genome-wide significant findings yet? Any known loci? If so, the authors should compare the results of their own study with previous findings (preferably in the Results section).

Results: The first section of “Results” is a cohort description which, to my opinion, belongs into the Methods. What does “characterizing” mean in this context?

Results: Are heritability estimates on liability or observed scale? This should be mentioned in the main text (and in the legend to Figure 1a).

Results: The authors state that “Heritability estimates [showed] comparable heritability estimates for clinical and self-reported traits”. Looking at Figure 1a this seems true for DKFM/DFSS, but not for periodontal and loose teeth, which are very different in their estimates despite a strong genetic correlation of 1 (see Figure 1b and major comment).

Results: the two pairs of overlap for analysis were DMFS and dentures, and periodontal disease and loose teeth, respectively. While the first one is highly significant, the second pair is provided with a P-value of 0.71 which, I presume, is a typo??

Results: For rs121908120, the variant with the largest effect size in the DMFS/dentures, the beta-value for the effect allele is <0, which I interpret as a protective effect of this minor allele? In turn would this mean that the common allele at this position is the risk allele at population level?

Results: The description of the findings at rs1122171 does not make sense to me. The authors state that this is a common variant near C5orf66, but then name PITX1, PCBD2 and CATSPER3 as genes that are spanned by the association signal. For this variant the authors check chromatin interactions and eqtl-data, however, they do not mention the limitations of this analysis (such as that the adequate tissue is not represented). Also, why have the chromatin / eqtl-interactions not been performed for other loci?

Results: many of the results (such as the strong association at the HLA region) seem to point to an immunological component. This is very interesting and could be highlighted / emphasized a bit more – also related to what was known before genetically.

Results: The annotation of the rs12461706-association with GTEx data and the presence of the SIGLEC5-mutation rs3829655 is a very interesting information. I am just wondering whether there is more or less expression for the risk allele?

Results: Gene set enrichment – how was the target gene / associated gene defined? It is more and more evident that the effect gene of associated variants is not necessarily the nearest gene, which is (currently) most often the gene considered for gene-based analyses. To get more biological insights, the authors might consider pathway analyses on their GWAS results.

Discussion: The first paragraph of the Discussion does not capture the essence of the paper, which I think it should do. Why don't the authors stress their approach again and highlight the fact that this study is the largest GWAS on dental traits with significant findings?

Discussion: What does “.. facilitating comparison with other traits” (page 18, 1st paragraph) mean?

In the Discussion the authors focus on WNT10A as candidate at one of the loci, which I think is valid given the evidence for involvement of this gene in tooth development and formation. However, there should be more information on where the association signal is located, how far this is from WNT10A coding sequence, and whether there is any evidence for interaction among the associated region and the gene.

The Discussion overall is very descriptive and a bit boring to read, there is redundancy and a focus on a number of technical details. This section might benefit from a more “lively” discussion of the impact, and a stress of highlights such as the immunological involvement / HLA region.

Methods: The dental disease trait descriptions in GLIDE could well be summarized in a Table.

Genome-wide analysis of dental caries and periodontitis combining clinical and self-reported data.

Dmitry Shungin, Simon Haworth, Kimon Divaris, Cary S. Agler, Yoichiro Kamatani, Myoung Keun Lee, Kelsey Grinde, George Hindy, Viivi Alaraudanjoki, Paula Pesonen, Alexander Teumer, Birte Holtfreter, Saori Sakaue, Jun Hirata, Yau-Hua Yu, Paul M. Ridker, Franco Giulianini, Daniel I. Chasman, Patrik KE. Magnusson, Takeaki Sudo, Yukinori Okada, Uwe Völker, Thomas Kocher, Vuokko Anttonen, Marja-Liisa Laitala, Marju Orho-Melander, Tamar Sofer, John R. Shaffer, Alexandre Vieira, Mary Marazita, Michiaki Kubo, Yasushi Furuichi, Kari E. North, Steve Offenbacher, Erik Ingelsson, Paul W. Franks, Nicholas J. Timpson, Ingegerd Johansson

Response letter format

This letter describes changes made in response to editorial feedback (first section), reviewer comments for the GWAS submission (second section) and reviewer comments for the previous standalone MR paper (third section). References and review table 1 follow section 3.

In each section, editorial and reviewer comments are in bold type. The response to these suggestions is demarcated by a double asterisk symbol and is provided in plain text. Sections from the revised manuscript are quoted using inverted commas and in italic font.

Since initial submission the term 'periodontal disease' has been replaced with 'periodontitis' to reflect recent changes in nomenclature¹. The reviewer comments have not been changed, however the response to comments and revised manuscript include the updated terminology.

Section 1 – Editorial comments

We had overlapping reviewers for both manuscripts and both reviewers felt that the MR study on its own was rather limited in scope and we fully agree with this. We also note that Reviewer #1 mentions a lack of biological insight from the GWAS study here and that MR would help to include more novel biological insight. In a revised version, we would ask that you ideally combine the two papers into a single manuscript and re-submit through the link below.

****As suggested the two papers have been combined for re-submission. The revised GWAS paper contains other changes to provide additional biological insight as described below in the response to reviewer 1.**

Section 2 – GWAS paper reviewer comments

Reviewer 1

This paper describes the largest study to date aiming to identify genetic determinants of Dental caries and periodontal disease. Whilst the sample size is impressive and many of the findings highly significant, I have a number of concerns with the manuscript:

1.1 The results section is very poorly written and structured, with many aspects of methodology unclear. For example, the number of genetic determinants is initially fixed at 17, but then changes or is re-assessed at half a dozen other occurrences throughout the results section. This leads to great confusion as to how many loci the authors are claiming and how robust the signals are. This inevitably leaves the reader navigating the huge number of poorly described supplementary tables in search of answers. I appreciate the complexity when trying to combine multiple traits, ancestries and analytical techniques, but the current approach is far from clear. I'd advise the authors consider what they feel is the most appropriate set of parameters and then present a single discovery analysis for the two traits.

****Following this advice the results section has been re-written. At the start of the single variant results section, two primary analyses are clearly defined. The number of genomic risk loci in each of**

these analyses are then defined using single set of criteria. We hope this approach provides greater clarity about the primary single-variant results.

1.1 (continued) Appropriate sensitivity analyses can then be performed but always on the same number of X variants. It seemed slightly illogical to present MTAG as secondary/sensitivity when it identified so many more variants than the primary analysis. The authors should come to a consensus view on what they feel is robust or not without just leaving it to reviewers to navigate the many supplementary tables.

**Following the suggestion above the strategy has been simplified. All single-trait, ancestry-specific and other sensitivity analysis are now used only to examine effect estimates at the lead single variants defined in the primary analysis. These are presented under a new results section entitled 'Population-specific sensitivity analyses'. To simplify the message and shorten the sensitivity analysis section the MTAG analysis has been removed. The section previously entitled 'secondary single-variant results' is no longer relevant and has been removed.

1.2 The number of defined independent signals in Table 1 looks concerning – presumably the multiple HLA signals are shaded in grey because the authors don't believe they're all truly independent? Perhaps the safest approach is to only report the most significant HLA association if this region can't be fine-mapped with more confidence.

** We have simplified the presentation of this table and now only report a single lead variant for the HLA region as suggested. The purpose of the grey shaded region in table 1 was to flag the presence of association signal within the HLA region. Given that the HLA region is reported to have variable LD structure² we are not confident that loci > 500kb apart represent independent signals of association and not confident that approximate conditional analysis using reference LD data can be used to clarify this. Instead, we have included analysis of imputed HLA haplotypes as discussed elsewhere in this response.

1.2 (continued) Can the authors also confirm that the remaining signals in that table are all uncorrelated with each other? (LD can range beyond 500kb).

**Where there is more than one risk locus on a chromosome, the signals included in table 2 (previously table 1) are conditionally-independent. This is tested using a GCTA stepwise selection procedure which was not included in the original submission. The methods section on identification of associated variants has been updated as follows;

"Genome-wide significance was set at $P < 5 \times 10^{-8}$. Within any given genomic locus, the lead SNP was defined as the variant with the lowest association p-value within a +/- 500 kb distance, using EasyStrata³. To test for conditional independence of lead variants, a stepwise selection procedure was performed using approximate conditional analysis⁴, implemented using the cojo-slct function of GCTA (version 1.91.4)⁵. Loci were only considered functionally independent if at least one variant reaching genome-wide significance was selected in that risk locus during this procedure. For the HLA region (chr6: 25-35 Mb) the broad association signal and complex LD structure prevented testing for conditional independence between variants, so only a single lead variant was reported for this region. Where more than one conditionally-independent variant was selected within a single risk locus, the lead variant was presented in Table 2, with additional variants presented in the supplementary material"

Methods, Pages 37-38

1.2 (continued) Was the associated HLA haplotype driving the association with any of the reported index variants or just coincidentally correlated?

**The haplotype analysis was undertaken using participant-level data in UKBiobank, while the meta-analysis combined summary level data from several studies. We are therefore unable to test SNP

association statistics after conditioning on haplotype, and have clarified this in the results section (page 13)

“To characterize this association signal, analysis of imputed HLA haplotypes was undertaken in UKB only (n=336,038) using HLA haplotype dosage as the exposure and ‘Dentures’ as the outcome “

1.3 The LDSC intercept of 0.81 for the DMFS/dentures discovery analysis suggests quite substantial over-correction of test statistics. On looking through the methods it appears that the UKBB results were corrected for a lambda GC of 1.46. This makes no sense at all considering the results have been generated from a linear mixed model and represents a severe over-correction. A more moderate approach would be to correct for the intercept value.

******The two primary meta-analyses have been updated including a correction term for the UK Biobank LDSC intercept terms rather than lambda GC. This is reflected in the updated methods (page 35-36) and in all results.

“The association statistics were corrected using the estimated intercept term from univariate LDSR⁶ (for dentures LDSR_i=1.0784, for loose teeth LDSR=1.0432) prior to meta-analysis with GLIDE.”

1.3 (continued) Can the authors confirm what ancestry groups were included in the UKBB analysis and whether this included related individuals?

******The UK Biobank analysis included related participants but was restricted to participants of European ancestry.

A k-means cluster analysis was performed using the first 4 genetic principal components. The single largest cluster (n=464,708) were included in analysis, representing participants of European ancestry. This information is included in the methods

Methods (Pages 33-34)

“Participant level QC involved restricting analysis to participants of European ancestry (identified as the largest cluster formed after k-means clustering on the first 4 genetic principal components, N=464,708) as well as standard exclusions for mismatch between reported and genetic sex, possible sex chromosome aneuploidy, outliers in heterozygosity and missing rates”

Two participants were removed because they were related to an implausibly large number of other study participants (>200). Other related participants who passed all genetic QC measures were included in analysis.

Methods (Page 34)

“An additional 2 participants were removed as they were apparently related to a very large number of participants at 3rd degree or closer; otherwise related participants passing all other quality control were included in single variant analysis⁷ “

1.4 What version of the UKBB imputation was used? If not v3 (which I’m guessing due to X-chromosome omission), this should be rerun as it’s likely to boost signal identification considerably.

******All single-variant analyses in UKBiobank as well as results from meta-analysis of Biobank and GLIDE have been updated using v3 of the UKBiobank imputation.

As well as inclusion of chromosome X in the primary results, this change means that simple structural variation with MAF > 1% is now included in analysis, totalling around 1 million additional variants.

This is reflected in changes throughout the paper including the methods (page 34)

“Single variant analysis used the 2018 (v3) imputed data release including imputation to both Haplotype Reference Consortium and UK10K/1000 genomes combined panel”

1.5) Integration with eQTL data has been performed using FUMA and S-PrediXcan, however to my knowledge neither are evaluating co-localisation of signal. For example, are many of the overlaps with expression driven by coincidental overlap of LD rather than truly shared effects? It would also be informative to show how many transcripts map to each of the individually associated regions.

******This is an excellent suggestion, though (especially with the inclusion of new data (see below)) we have to consider the primary objectives of the paper and the bounds of this specific investigation. The aim of this paper was to test for the presence of genetic association signal for the major dental disease traits and share hypothesis-generating research data with the research community. Given the lack of functional follow-up experiments we do not claim to have fine-mapped any signals to their biologically causal gene. We will look to do this in the future.

At a single-variant level we acknowledge that the annotation of a single variant with an eQTL may be coincidental and have therefore removed any statements suggesting that these eQTL overlaps point to a biologically causal gene.

At a gene-based level, the S-PrediXcan approach accounts for correlation between SNPs when training prediction models through use of LASSO and elastic net training procedures, meaning that SNPs which are correlated through LD are either down-weighted or excluded from the model. We feel this approach therefore allows robust inference at a locus and transcript level, but it is not intended to provide inference at a single-variant level. We have removed any statements which imply inference about a lead SNP was enhanced by S-PrediXcan analysis

1.6) Many supplementary tables contain variant names which are cryptic and uninformative – this should be clarified.

******All supplementary tables and supplementary data have been updated. Gene names are now provided using HGNC nomenclature.

1.7) Biological insight seems somewhat limited. The only pathway approach run is DEPICT, which didn't identify any pathways. Have the authors considered using other approaches?

******Following reviewer and editorial feedback the follow-up analyses have been comprehensively revised, within the bounds and scope of this paper as described in response to comment 1.5.

In brief, this includes

- a) Running additional gene set and pathway analysis
- b) LDSC stratified by genomic annotation
- c) LDSC partitioned by cell type.
- d) Predictions using S-PrediXcan have been updated to take advantage of a recent extension to this method which accounts for cross-tissue correlation and gains statistical power.
- e) Systematic screen for genetic correlation through hypothesis-free cross-trait LDSR
- f) An illustrative example of Mendelian randomization

We hope these changes, which are described in detail in subsequent responses, provide additional biological insight.

1.8) What cell and tissue types were significantly enriched (most readily tested using LDSC-SEG)? This may be informative for downstream eQTL testing.

******As suggested, we have included stratified LDSC which is included in the following passages of the manuscript;

Results (Pages 12-13)

“Stratified LDSR was performed to test whether the heritability of dental disease traits was enriched functional elements of the human genome⁸ or in regions surrounding genes with tissue-specific expression patterns⁹. In the DMFS/dentures combined analysis there was evidence for enrichment in 38/85 functional annotations including annotations providing evidence for evolutionary conservation. Treating Genomic Evolutionary Rate Profiling (GERP) scores as a continuous annotation, DMFS/dentures association signal was enriched in genomic regions under evolutionary constraint ($p=1.5 \times 10^{-28}$), while the 1.9% of variants annotated as highly conserved from primates to humans accounted for 30% of the heritability of DMFS/dentures (15.5-fold enrichment over baseline, $p=3.0 \times 10^{-15}$; Supplementary Data 9). In LDSR analyses stratified by tissue annotation, the highest LDSR coefficients were seen for genomic regions specifically expressed in gastrointestinal tract and minor salivary gland tissue; however, coefficients were imprecisely-estimated, preventing tests for difference between tissue types (Supplementary Data 10). In the periodontitis/loose teeth analysis, there was little evidence for enrichment in functional or tissue annotation groups beyond chance (Supplementary Data 11, Supplementary Data 12), possibly reflecting limited statistical power for these analyses.”

Discussion (Page 17)

“while integration with functional annotation data showed that heritability of dental caries was enriched in conserved genomic regions, with a fold enrichment value similar to other complex traits with serious health consequences⁸”

1.9) “Single-variant cross-trait lookup” – information in this paragraph very limited and points to verbose supplementary tables. Key information is what % of identified signals overlaps a non-oral health related complex trait.

******A summary of this information is now included in the results text as and as a summary table, with extended content included as Supplementary Data.

Results (Page 10)

“Using the PhenoScanner browser¹⁰ (see Methods) to make comparison with published studies, 46/47 variants or proxies ($r^2 > 0.8$, 1KGP phase 3, European ancestry reference data) were available in the PhenoScanner database, and 26 of these (57%) have previously-reported association with one or more non-dental diseases or health traits, termed ‘overlaps’ in this description. Traits seen repeatedly in this overlap analysis include adiposity traits (11/26 overlaps), height (9/26 overlaps), and bone mineral density (3/26 overlaps). A summary is shown in Supplementary Table 4, with full results for each variant in Supplementary Data 11.”

1.10) Genetic correlations without follow up Mendelian Randomization are of limited value and likely mirror what you might observe in an epidemiological observational setting. At a minimum it would be informative to test the educational attainment and smoking bi-directional MRs in this manuscript.

******We accept that genetic correlations may recapitulate those seen in a conventional epidemiological study, but believe they are still valuable for understanding the composition of complex phenotypes, and in flagging risk factors or diseases which may be tied to the phenotype through causal or pleiotropic pathways rather than confounding alone.

To maximize value from this part of the paper we have updated the genetic correlation analysis and now perform this in a hypothesis-free manner, identifying 43 diseases or traits with evidence for non-zero correlation with dental diseases.

The limited statistical power for MR experiments treating dental disease as an exposure means it is necessary to reduce the multiple testing burden by selecting a subset of traits. As a concise example for the paper we have focused only on cardiometabolic diseases and risk factors because

- a) there is good observational and mechanistic evidence to suspect causality
- b) there is evidence for genetic correlation for example between DMFS/dentures and coronary artery disease
- c) there is conflicting interpretation in the literature with insufficient evidence from clinical trials to support or refute causality
- d) the aim of the MR analysis is to illustrate value in these GWAS data, not as an exhaustive investigation

We feel this combination of features means a cluster of phenotypes surrounding cardiometabolic health is the most compelling illustration for the present paper. We accept this investigation is relatively narrow in scope but will of course make these data available in their entirety for other researchers to examine other research questions.

1.11) Table 2 contains 3 data points and may be better served integrating into Table 1.

** We accept the need to simplify the presentation and have done this by removing table 2, presenting relevant information in the text and focusing on the presentation of novel signals.

The motivation for this decision (rather than merging with another table) is that the contents of table 2 (previously table 1) are all novel GWAS signals sharing the same units for the same trait, while the contents of previous table 2 represent a replicated GWAS signal with different units and for a different trait.

Reviewer 3

In the paper under review, the authors perform a large-scale GWAS/meta-analysis on two dental traits, caries and periodontitis. The sample used for this analysis was composed of a well-characterized consortium project GLIDE, which provided extensive clinical data, and a more population-based and self-reported data set from the UK Biobank. With their study the authors suggest a possible approach to answer a very important question in complex genetics research: How can those large sample sizes be achieved that are required for the detection of low effect sizes? They test the combination of diverse measures as proxy for one specific trait, identify those pairs that provide the best genetic correlation, and maximize the sample size using these two measures from both cohorts. The approach worked very well for caries, while results are less clear / significant for periodontitis (see my comments).

**Following substantial changes to the paper we now place greater emphasis on results for DMFS/dentures and less weight on those for periodontitis/loose teeth. Please see a point by point responds for detailed comments.

The authors well controlled for any potential biases when combining both data sets, the results (at least for dental caries) seem to be conclusive and reliable. However, it would be very interesting to the reader to see whether the approach of combining clinical samples and large-scale (less well characterized) samples has been previously performed by other groups / for other traits?

**Thank you for this comment. We are not aware of this approach being applied for dental disease traits. For other complex traits we are aware of one recent example where family history of Alzheimer's disease was used as a proxy for clinical diagnosis in meta-analysis, validated by assessing genetic correlation. We have included this item in the discussion (Page 17).

"This investigation used an empirical approach informed by shared heritability to select dental disease proxy measures with similar biological meaning to detailed clinical measures. This approach

has similarities to a recent investigation which used genetic correlation for post-hoc confirmation of a prior expectation that family history of Alzheimer's disease is a satisfactory proxy of clinical diagnosis¹¹, but here we extend this approach to the selection of disease proxies where the prior expectation is less clear. This workflow may be adaptable to other complex diseases where it is difficult to obtain detailed phenotypic information in large collections"

I have identified some critical points that mainly relate to some details of the analyses and restructuring of the results section. As major benefit, the paper under review generates the possibility for a lot of follow-up hypotheses in order to understand the biology underlying dental traits.

****Thank you for this comment. The value of GWAS results as a resource for the research community is now included as a discussion point (Page 18).**

"These genetic associations improve disease understanding and may provide a foundation for innovative approaches to risk assessment, outcome prediction, disease management or treatment. The availability of genetic proxies for dental diseases in large datasets also unlocks previously-unavailable approaches such as two-sample MR¹² to infer potential consequences of dental diseases. As an illustrative example, the MR analysis included in this work provides some support for the hypothesis that dental diseases proxied by DMFS/dentures may be an upstream risk factor for metabolic disturbance and cardiovascular disease events and prioritizes specific relationships for additional investigation. This analysis also provides a broader demonstration of the utility of these data for generating and testing research hypotheses."

3.1) Analysis I: For the DMFS/DFSS analyses the authors included about 25.000 samples from several studies of which one (one of the smaller ones) was Non-European. This inclusion complicated the analyses (as two meta-analyses were done – one for Europeans, and one for "mixed ethnicities" with the contribution of the one additional sample). Similarly, two ethnicity-analyses were done for periodontal disease. Given the results I am not sure that this inclusion made sense – from about 489,000 individuals included in the DMFS/dentures analysis, only about 20,000 were from Hispanics.

****We have simplified the analytical strategy for periodontitis in line with suggestions made by reviewer 1. We now report a primary analysis in participants of European ancestry. Following this, there is a lookup in independent meta-analysis of East Asian ancestry participants. We feel this is the best approach to satisfy the request for a clear primary analysis.**

For DMFS we have chosen to present a primary analysis including a study of Hispanic/Latino ancestry, with ancestry-specific effect sizes as a sensitivity analysis. We feel this is justified as;

- a) this study makes a large contribution to the available clinical samples in GLIDE for DMFS (around 44%)
- b) there is no evidence for heterogeneity in genetic effect sizes in this study compared to other studies in GLIDE ($P_{FDR} > 0.05$ for all variants)(Supplementary Table 6)
- c) the patterns of LD decay (and therefore LD scores) in admixed Hispanic/Latino populations are broadly comparable to patterns in European populations (Figure 4 in¹³)

3.1 (continued) As a consequence of this, the reader gets confused at several points as to which analysis the authors talk about. For instance, in the LDSR analyses (third part of the Results section), I assume it is European samples only? Or does this refer to the combined samples? As another example, page 11 (attenuation of the effect of rs12461706), it just becomes clear at the end of this page that all other analyses before were performed without the East Asians. I think that many of this information is provided in the Methods, but it would be very beneficial to have this included in the main text also (or, alternatively, to leave out the "mixed ethnicity" analyses).

**As discussed in the response to comment 1.1 there is now a clearly defined primary analysis for two GLIDE+UKB combinations. Results from these primary analyses populate all follow-up analyses, and sensitivity analyses are now only performed on variants which are considered lead signals in the main analysis. We hope that this revision addresses potential confusion in the previous manuscript. The LDSR analysis included results of the HCHS/SOL study. Estimates of heritability and shared heritability with dentures were similar when this study was excluded or included, but with greater precision in the estimates when the larger sample size was included (Review Table 1). This has been clarified in the methods (pages 30-31)

"In 9 studies, analysis was conducted in individuals of European ancestry (ARIC, COHRA1, DRDR, MDC, NFBC1966, SHIP, SHIP-TREND, TWINGENE, WGHS). In one study (HCHS/SOL) participants were recruited from Hispanic and Latino communities in the USA, who self-reported ancestry from 6 broad groups (Cuban, Dominican, Mexican, Puerto Rican, Central American, South American). To undertake analyses within this highly admixed population a bespoke modelling approach was undertaken. Multi-dimensional clustering was used to generate genetic analysis groups containing participants of similar ancestry. These group allocations were then used as covariates in a linear mixed model (partitioned to only fit the proportion of genetic structure due to familial relatedness rather than ancestry) alongside the first 5 genetic principal components, study center and log-transformed sampling weights¹⁴. Subsequently, the results from HCHS/SOL were treated as a study of European ancestry and included in the primary meta-analysis."

3.2 Analysis II: As seen in Figure 1, the genetic correlation between loose teeth (UKB) and periodontitis (GLIDE) is 1. However, the heritability estimates provided in Figure 1a show significant differences between these two traits. This leaves me questioning whether there is a statistical error in some of these analyses, or what the explanation for this discrepancy is. Given these differences in heritability estimates, is the combination of these two traits in a meta-analysis justified?

**As mentioned in the discussion section, the estimated heritability of periodontitis was low, meaning that results downstream of this should be interpreted with caution. Nevertheless, we believe that the combination of periodontitis and loose teeth is justified on this occasion because;

a) There is sound prior clinical rationale.

Persistent loose teeth are caused by advanced periodontitis. Dental trauma may cause loose teeth, but this is uncommon in middle aged and older adults, likely to be transient in nature and not correlate with genotype. Thus, there is a prior clinical expectation that loose teeth should act as a proxy for periodontitis.

b) We used an appropriate meta-analysis approach

The approach used to combine these traits takes the evidence that a SNP has a non-zero effect on one trait (expressed as a Z score) and combined that with the Z score for the other trait. It is not necessary for the genetic effects to be on the same scale, meaning that differences in the heritability of the two traits should not bias the interpretation of genetic correlation estimates.

c) Post-hoc, the empirical findings have face validity

This analysis confirmed association at a known risk locus in the absence of major inflationary bias, despite association not being seen in either of the contributing single-trait analyses. This seems to support the meta-analysis being a valid manoeuvre.

The reviewer queries whether large R_g values should be observed for traits with low heritability. This is possible as the test examines whether genetic effect estimates are correlated in size and direction of effect. There is no requirement that effect estimates are on the same scale. Thus, a trait with very low heritability can have an R_g value of 1 with a trait with very high heritability provided the

detectable genetic effects for the trait with poor heritability are perfectly correlated in size and direction with those for the other trait.

Finally, although we believe the inclusion of periodontitis and loose teeth in combined analysis is justified, we accept the findings are less convincing than for dmfs/dentures and have adjusted the space allocated to these in the manuscript accordingly.

3.3 Results/conditional analyses: The authors defined independent effects as variants that, after conditioning on the sentinel SNP, reached genome-wide significance, and which are located outside a 500 kb region to either side. This approach is of high risk for false-negatives, as independent effects might be present within the 1 Mb around the lead SNP. What is the rationale for the 500 kb window? Also, it is questionable whether independent effects have to require genome-wide significance, given the limited power as compared to the first analysis – a decrease in P-values might be a good indicator, too. The authors should consider to re-run the conditional analyses and consider other organizational elements of the genome such as TADs for definition of possible boundaries.

******The main aim of the single variant results is to show novel and conditionally-independent risk loci. We have updated the criteria for table 2 to make this clearer, and explicitly test conditional independence in defining these loci as described in the response to comment 1.2.

This procedure highlights 3 loci containing multiple signals of association – these are now flagged in the main results (page 9)

“For DMFS/dentures there were 47 conditionally-independent genetic risk loci where at least one variant met a conventional threshold for genome-wide significance ($P < 5 \times 10^{-8}$) in the unconditional results and following a stepwise selection procedure (Table 2; Fig. 2a; Methods). The approximate conditional analysis procedure (Methods) also highlighted 3 risk loci containing multiple conditionally-independent signals of association (Supplementary Table 3).”

Minor comments:

3.4 Abstract: The abstract would benefit from more details on the results – for instance, some highlights from the genetic correlation analysis or some of the interesting loci from the caries association study. At the moment the Abstract is more a project description and not very attractive to the reader.

******Following major changes to the primary results and follow-up analyses the abstract has now been revised to focus on the main findings.

3.5 Introduction: There is a number of superficial statements in the Introduction. Some examples: “Caries is the most prevalent disease worldwide” – this is hard to believe, and I was not able to find proof for this statement in the provided reference. How frequent is it (is it truly more prevalent than, for instance, cancer)? What is the evidence for this statement?

******In the 2016 global burden of disease study¹⁵ study dental caries in permanent teeth was estimated to be the most prevalent disease (affecting 2.4 billion adults at any moment in time)(with second-highest incidence after upper respiratory infections (7.3 billion incident cases in 2016, reflecting the high recurrence rate of caries after treatment). We feel the statement made in the introduction is justified, but have clarified the source of the prevalence estimates in the introduction (Page 4)

“The 2016 Global Burden of Diseases, Injuries, and Risk Factors Study¹⁵ estimated that dental caries in permanent teeth and periodontitis were the leading and 11th most prevalent causes of disease worldwide in 2016”

3.6 As another example, the authors state that evidence for a heritable contribution exist “along with an obvious environmental burden”. What does “obvious” relate to? How much is the heritable estimates? Where do the numbers come from, and what is the nature of the environmental factors?

**Both dental caries and periodontitis have reported heritability estimates of up to 50%. This figure is now included in the manuscript along with references to 2 original research articles and a comprehensive review article which explores this further. Introduction (page 5)

“Part of this story is the genetic contribution to oral health outcomes and the heritability of dental caries and periodontitis have been reported to be as high as 50%¹⁶⁻¹⁸.”

The best-known environmental risk factors for dental diseases are diet, oral hygiene and fluoride intake, but many others have been postulated. The focus of this paper is on genetic risk for dental diseases, therefore comments on environmental risk have been removed from the introduction along with other superficial statements.

3.6 (continued) Finally, referring to previous studies, the authors stated that they had “yielded interesting results”. What is known about the traits? Any genome-wide significant findings yet? Any known loci?

** This passage of the introduction was intended to highlight the discrepancy between the enormous burden of dental diseases and lack of any well-powered consortium-based GWAS.

We considered whether to expand the introduction to include a more comprehensive review of previous studies and candidates. This challenging given both space requirements for this manuscript and the complexity of different approaches used to disease classification. We have therefore included reference to a relevant review article.

For dental caries (Introduction, page 5)

“To date few reliable association signals have been found and this is likely to reflect different measurement approaches used, complex genetic architecture¹⁹ of dental caries or limited statistical power to detect associations”

For periodontitis (introduction, page 5)

“These studies have not yielded consistent evidence of specific genetic contributions to periodontitis.²⁰”

3.6 (Continued) If so, the authors should compare the results of their own study with previous findings (preferably in the Results section).

**As suggested we have amended the results section to clarify which findings are considered novel and which are considered positive controls.

Results (Page 10)

“The remaining 45 genetic risk loci in the DMFS/dentures combined analysis include some investigated in candidate gene studies for dental caries such as the Human Leukocyte Antigen (HLA) region²¹, but none which have previously been identified directly in other GWAS for dental diseases.”

Results (Page 10)

“In the periodontitis/loose teeth combined analysis there was evidence for association at a single risk locus, SIGLEC5, which was recently reported as a risk locus for aggressive periodontitis²²”

3.7 Results: The first section of “Results” is a cohort description which, to my opinion, belongs into the Methods. What does “characterizing” mean in this context?

**This section has been shortened and some passages moved to the Methods as suggested. The word ‘characterizing’ has been replaced with ‘genome-wide analysis of’.

3.8 Results: Are heritability estimates on liability or observed scale? This should be mentioned in the main text (and in the legend to Figure 1a).

**The study participants were not recruited on the basis of their dental disease status. Disease prevalence in the study group population is expected to reflect the underlying population, meaning that the liability and observed scale estimates are equivalent.

3.9 Results: The authors state that “Heritability estimates [showed] comparable heritability estimates for clinical and self-reported traits”. Looking at Figure 1a this seems true for DKFM/DFSS, but not for periodontal and loose teeth, which are very different in their estimates despite a strong genetic correlation of 1 (see Figure 1b and major comment).

**This statement was intended to mean that, as a group of traits, self-reported traits had similar heritability estimates to clinically assessed traits. The statement was not intended to mean that all traits had similar heritability estimates, and has been replaced with a less ambiguous statement

Results (Page 7)

“In GLIDE, the highest and lowest heritability estimates were reported for Nteeth (0.13, se=0.02) and periodontitis (0.01, se=0.01) respectively, while in UKB the highest and lowest heritability estimates were for dentures (0.09, se=0.004) and toothache (0.04, se=0.007) respectively (Fig. 1a) ”

3.10 Results: the two pairs of overlap for analysis were DMFS and dentures, and periodontal disease and loose teeth, respectively. While the first one is highly significant, the second pair is provided with a P-value of 0.71 which, I presume, is a typo??

**The low heritability of periodontitis means that the standard errors of the R_g estimate is very wide and the p value is large. The inclusion of $P=0.71$ in the text was intended to flag this to the reader so they know to interpret the R_g value with caution. Following this suggestion this point is now made explicitly in text. Results (Page 8)

“For periodontitis, the highest R_g value was seen for loose teeth ($R_g \sim 1$) however this was imprecisely estimated due to the low heritability of periodontitis (se=0.78, p=0.17).”

3.11 Results: For rs121908120, the variant with the largest effect size in the DMFS/dentures, the beta-value for the effect allele is <0, which I interpret as a protective effect of this minor allele? In turn would this mean that the common allele at this position is the risk allele at population level?

**This interpretation is correct. To preserve concordance between raw results files and tables we have avoided transposing allele frequencies or effects wherever possible.

3.12 Results: The description of the findings at rs1122171 does not make sense to me. The authors state that this is a common variant near C5orf66, but then name PITX1, PCBD2 and CATSPER3 as genes that are spanned by the association signal. For this variant the authors check chromatin interactions and eqtl-data, however, they do not mention the limitations of this analysis (such as that the adequate tissue is not represented). Also, why have the chromatin / eqtl-interactions not been performed for other loci?

**We are aware of the major limitations of gene mapping based on position alone. We aimed to supplement this with functional information from chromatin interactions and eQTL data. This approach identifies several plausible candidate genes at most loci (and a total of 1,048 potential

candidate genes genome-wide) which it was not possible to discuss for all loci, so only a lead signal was discussed in the previous draft.

We acknowledge this approach was not well communicated, leading to confusion about the description of this locus.

We have now simplified the main messages that a) there is evidence for association at this locus and b) we have not mapped this to a biologically functional gene.

Results (Page 10)

“The lead variant lies within an uncharacterized protein-coding region, C5orf66, however putative gene mapping using reference expression quantitative trait locus (eQTL) and chromatin interaction data implemented in FUMA²³ highlighted several potential genes at this locus.”

As suggested, the lack of availability of expression data in relevant tissues has been flagged as a limitation of these analyses in the discussion

Discussion (Page 19)

“While these approaches maximize value from GWAS and improve statistical power to identify relevant genes²⁴, they are limited by the quality and availability of external data, for example data on odontoblast or ameloblast tissue are not available in GTEx.”

3.13 Results: many of the results (such as the strong association at the HLA region) seem to point to an immunological component. This is very interesting and could be highlighted / emphasized a bit more – also related to what was known before genetically.

**The HLA region has been investigated as a candidate gene for dental caries in the past, and variation at this locus may influence the oral microbiome. This is mentioned in the results (Page 13)

*“HLA class II molecules are expressed by antigen presenting cells and alleles of HLA class II are thought to modulate the composition of the oral microbiome, including the cariogenic gram-positive organism *Streptococcus mutans*²⁵”*

For other loci we have tried to use a systematic approach to highlighting findings based on the strongest single-variant evidence for association, largest single-variant effect estimate and strongest evidence for association in the transcriptome-wide association study part. There is therefore limited opportunity to emphasize immunological loci, but we have included a note that the carbonic anhydrase family may regulate the oral microbiome alongside other plausible mechanisms (Page 11)

*“Defects in CA12 lead to impaired salivary function and xerostomia²⁶, which is a risk factor for dental caries, while other members of the carbonic anhydrase family including CA6 may also play a role in regulating tooth biofilm microbiota and colonization by the cariogenic microorganism *Streptococcus mutans*²⁷”*

3.14 Results: The annotation of the rs12461706-association with GTEx data and the presence of the SIGLEC5-mutation rs3829655 is a very interesting information. I am just wondering whether there is more or less expression for the risk allele?

**The risk-associated T allele of rs12461076 is associated with increased expression of SIGLEC5 in all tissues where the eQTL association is considered significant by FUMA. However, in response to the need to shorten discussion of periodontitis findings and the comment that eQTL overlap might be coincidental, we have now removed this part from the updated manuscript.

3.15 Results: Gene set enrichment – how was the target gene / associated gene defined? It is more and more evident that the effect gene of associated variants is not necessarily the nearest gene,

which is (currently) most often the gene considered for gene-based analyses. To get more biological insights, the authors might consider pathway analyses on their GWAS results.

******The gene set enrichment analysis strategy has been expanded following the advice of reviewer 1, and now includes a complementary approach implemented in FUMA

DEPICT defines association signal at a locus level, where a locus can contain several genes. Although this is position-based, DEPICT can account for the associated gene not being the closest to the lead signal, provided it is in the same locus.

The approach implemented in FUMA uses the results of the S-PrediXcan analysis which integrates eQTL data with GWAS results to prioritize a list of candidate genes and is therefore more likely to capture relevant target genes than an approach based on position alone. This approach uses the MSigDB C2 gene sets, which includes 1329 canonical pathways.

We view these approaches as complementary, and have updated the results and methods sections as follows;

Results (page 12)

“Gene set enrichment was assessed using DEPICT²⁸, which tests whether loci housing dental-disease-associated single variants are over-represented in predefined sets of functionally related genes. For the DMFS/dentures analysis there was no evidence for enrichment in gene sets passing an acceptable false discovery rate (Supplementary Data 6), while for the periodontitis/loose teeth analysis there were insufficient independently-associated loci to use this method. An alternative approach was implemented in FUMA²³ using a hypergeometric test to test whether the transcripts prioritized by S-TissueXcan were over-represented in curated gene sets defined in the Molecular Signatures Database (MSigDB C2). No enrichment beyond chance was observed after correction for multiple testing for DMFS/dentures loci, and there were insufficient loci to perform this analysis for periodontitis/loose teeth.”

Methods (page 39)

“DEPICT²⁸ (version 1.1, release 194), was used to test whether associated loci are preferentially located within pre-defined functionally similar gene sets, assessed against a null expectation informed by GWAS results for randomly distributed phenotypes. Associated loci were defined at ($p < 5 \times 10^{-8}$) in combined analysis. Reconstituted genesets, tissue expression matrix and gene annotation files were downloaded from the DEPICT repository (data.broadinstitute.org/mgp/depict/documentation).

A hypergeometric test for gene set enrichment was performed using FUMA²³. Associated genes were defined as those passing Bonferroni correction in S-TissueXcan analysis. Pathways and gene sets were taken from the Molecular Signatures Database (MSigDB C2), and all protein coding genes in the FUMA database were used as background. Enrichment p values were corrected for multiple testing using Bonferroni correction.”

3.16 Discussion: The first paragraph of the Discussion does not capture the essence of the paper, which I think it should do. Why don't the authors stress their approach again and highlight the fact that this study is the largest GWAS on dental traits with significant findings?

******The discussion now opens with a stronger statement describing the paper which captures the approach and main findings.

Discussion (page 17)

“This investigation used detailed clinical measures in combination with genetically-validated proxy phenotypes to investigate the major dental disease of caries and periodontitis. Combined analysis of

DMFS and dentures identified 47 novel risk loci, including common variation near PITX1, CA12 and in the HLA region in addition to uncommon variation with large effects in WNT10A. Combined analysis of periodontitis and loose teeth (a late symptom of advanced periodontitis) confirmed association at a previously reported locus, SIGLEC5. Integration of GWAS results with external sources of expression data prioritized 221 gene transcripts which may be involved in the pathogenesis of dental caries, while integration with functional annotation data showed that heritability of dental caries was enriched in conserved genomic regions, with a fold enrichment value similar to other complex traits with serious health consequences⁸. Cross-trait comparisons showed overlap in the genetic determinants of dental diseases and complex non-oral health traits at both a single variant level and genome-wide level. Finally, using a Mendelian randomization analysis framework there was evidence suggesting that biological processes leading to dental caries may have downstream effects on general health”.

3.17 Discussion: What does “.. facilitating comparison with other traits” (page 18, 1st paragraph) mean?

******This statement was meant to convey that generating genome-wide data for dental diseases is needed to unlock applied epidemiological analyses such as genetic correlation and Mendelian randomization, and is therefore an important step in understanding how dental diseases relate to other health traits. This argument is now made through illustrative examples and a specific statement about the value of GWAS findings (see opening comments in response to reviewer 3, Discussion, page 18)

3.18 In the Discussion the authors focus on WNT10A as candidate at one of the loci, which I think is valid given the evidence for involvement of this gene in tooth development and formation. However, there should be more information on where the association signal is located, how far this is from WNT10A coding sequence, and whether there is any evidence for interaction among the associated region and the gene.

******This variant is a mis-sense coding variant resulting in a phenylalanine > isoleucine substitution with predicted deleterious consequences. This information may not have been clear, so is now presented at when the *WNT10A* locus is first introduced (Results, Page 9)

“rs121908120 is a missense variant within WNT10A which results in a phenylalanine to isoleucine substitution and is predicted to have deleterious consequences in multiple WNT10A transcripts using the ExAC browser²⁹.”

3.19 The Discussion overall is very descriptive and a bit boring to read, there is redundancy and a focus on a number of technical details. This section might benefit from a more “lively” discussion of the impact, and a stress of highlights such as the immunological involvement / HLA region.

A new discussion section has been written to emphasise the highlights and potential impact of findings. The main changes are

- a) A clearer and stronger opening passage summarizing the approach and lead findings,
- b) Less emphasis on null / negative findings
- c) Greater emphasis on the possible applications and impact of findings
- d) A more concise summary of major limitations

3.20 Methods: The dental disease trait descriptions in GLIDE could well be summarized in a Table.

******A table has been created as suggested (Table 1).

Section 3 – MR paper

Reviewer 1.

MR1.1) Given the sole focus of this paper is MR and the GSMR method is relatively new, it would seem prudent to compare analyses using other available methods (for example, MR-PRESSO, IVW, ...). How robust are these observations when multiple techniques for association testing and heterogeneity assessment are used?

****Causal effect estimates from 3 other methods have been included as a supplement.**

MR 1.2) How did the authors deal with significant loci in the HLA region? This looked problematic in the GWAS paper and I'd recommend excluding this region if not done so already.

****For estimation of causal effects of DMFS/dentures the HLA region has been excluded apart from a single lead variant. This is clarified in the methods section (Page 41);**

"In the primary analysis, variants in the HLA region (chr6: 25-35 Mb) were removed for estimation of casual effects of DMFS/dentures, except for a single lead variant. For estimation of causal effects of other traits on DMFS/dentures, full genome-wide data were used"

MR 1.3) The paper would benefit from speculation about the mechanisms linking exposures to outcomes. In several cases the observed associations seem slightly far-fetched, so any supporting evidence would be insightful.

****While there are plausible potential mechanisms involving inflammatory mediators or a role of the oral microbiome it is not possible to review these in the limited space available within the GWAS paper.**

MR 1.4) Given many of these relationships are bi-directional, how can the authors rule out "common soil" effects where a single trait X is shared between the exposure and outcome? This seems potentially plausible in the context of traits such as cognition and personality. Related to this, can the authors perform multi-variate MR analyses to confirm that all the observed causal associations are independent?

****The "common soil" hypothesis is similar to MR comment 3.2 and discussed later in this response. The aim of the MR analysis is to estimate the main effects of a change in dental disease burden on population cardiometabolic health – while it would of course be interesting to also explore mediation and conditional-independence of causal effects, this falls outside the scope of the current manuscript.**

MR 1.5) What was the rationale for the selection of traits to be tested here? Why not educational attainment and smoking which had the strongest genetic correlations in the linked GWAS paper? Why not other obvious candidates of poor dental health such as alcohol consumption?

****The rationale for choosing these traits is given in response 1.10 earlier in this letter.**

Reviewer 3

The main focus of this paper is the estimation of causal relationships using Mendelian Randomization, a technique that has gained increasing attention in the community. However, there is also considerable criticism. Generally, it is of benefit to the society to understand causal relationships between adverse outcomes and other traits that are potentially actionable. This is also true for dental traits such as those analyzed here (although, actually, only one of two traits has been thoroughly investigated), and this analysis is the first of its kind for dental caries. My major concern relates to the selection of traits that were analyzed, which is limited to a set of nine, only two of which are cardiometabolic. It is unclear to me how the traits were selected. I

understand that BMI and fasting glucose were genetically correlated in the back-to-back GWAS paper, but there have been genetic correlations with other traits also, and these have not been included.

**The rationale for selecting traits has been described earlier in this response.

MR 3.1) Only GLIDE data were used. Why weren't UKB data included? The GWAS-paper has both data sets incorporated.

**All MR results use the combined analysis of GLIDE and UK Biobank. We hope this will be clearer now that the results are positioned in the GWAS manuscript after description of the combined analysis.

MR 3.2) A recent paper by O'Connor & Price (Nature Genetics, 2018) should be addresses / incorporated; it is a critics for false positives in Mendelian Randomization.

**This article is now cited in a short discussion about the potential challenges of MR for complex traits with partially overlapping genetic determinants.

Discussion (Page 20)

"We note that the inference drawn from all MR experiments will depend on the approach used to detect and account for horizontal pleiotropy³⁰ and shared latent genetic aetiology³¹. This is particularly relevant to complex traits such as dental caries, and several of the lead variants identified in the present study are in known risk loci for complex traits such as BMI. Non-specificity of the DMFS/dentures phenotype may also be relevant as downstream effects of DMFS/dentures could involve a mechanism of periodontitis"

MR 3.3) Abstract: The first sentence implies that both dental caries and periodontal disease will be analyzed in this paper, but this is not the case (periodontal disease is not analyzed in both directions due to the lack of genome-wide significant findings in the GWAS). The statement that a "series of traits" will be analyzed is misleading, as only 9 traits are selected. The second sentence puts emphasis on cardiometabolic disease, but this – in the end – is negative. This should be incorporated in the Abstract.

**Thank you for these comments. There is no longer a standalone MR abstract but we have been careful to avoid using the term 'series of traits' in the GWAS paper.

MR 3.4) The number of SNPs from dental traits to the others is very limited (about 20). Why wasn't a polygenic component (such as including more SNPs with higher P-values) analyzed?

**The decision to only include variants with $p < 5 \times 10^{-8}$ is based on a specific recommendation by the authors of the GSMR method.

*"Lastly, we have shown in a previous study that the SMR test-statistic is slightly deflated due to the use of a Taylor series expansion to compute an approximated sampling variance based on summary data, especially if the association between the instrument and risk factor is not strong enough. We therefore strongly recommend that only SNPs that are associated with the exposure at a genome-wide significance level (i.e., 5×10^{-8}) should be used in GSMR analysis, and as a rule of thumb advise application only when there are 10 or more independent (e.g., $r^2 < 0.05$) genome-wide significant SNPs."*³²

However, as there is now improved power in the main combined analysis of GLIDE and UKBiobank we have been able to increase the number of instruments without resorting to less-stringent p-value threshold.

References

1. Caton, J.G. *et al.* A new classification scheme for periodontal and peri-implant diseases and conditions – Introduction and key changes from the 1999 classification. **45**, S1-S8 (2018).
2. Miretti, M.M. *et al.* A high-resolution linkage-disequilibrium map of the human major histocompatibility complex and first generation of tag single-nucleotide polymorphisms. *American journal of human genetics* **76**, 634-646 (2005).
3. Winkler, T.W. *et al.* EasyStrata: evaluation and visualization of stratified genome-wide association meta-analysis data. *Bioinformatics* **31**, 259-261 (2015).
4. Yang, J. *et al.* Conditional and joint multiple-SNP analysis of GWAS summary statistics identifies additional variants influencing complex traits. *Nature Genetics* **44**, 369 (2012).
5. Yang, J.A., Lee, S.H., Goddard, M.E. & Visscher, P.M. GCTA: A Tool for Genome-wide Complex Trait Analysis. *American Journal of Human Genetics* **88**, 76-82 (2011).
6. Bulik-Sullivan, B.K. *et al.* LD Score regression distinguishes confounding from polygenicity in genome-wide association studies. *Nature Genetics* **47**, 291+ (2015).
7. Mitchell, R.E. *et al.* UK Biobank Genetic Data : MRC IEU Quality Control, version 2. (University of Bristol, Bristol, UK, 2019).
8. Finucane, H.K. *et al.* Partitioning heritability by functional annotation using genome-wide association summary statistics. *Nature Genetics* **47**, 1228 (2015).
9. Finucane, H.K. *et al.* Heritability enrichment of specifically expressed genes identifies disease-relevant tissues and cell types. *Nature Genetics* **50**, 621-629 (2018).
10. Staley, J.R. *et al.* PhenoScanner: a database of human genotype–phenotype associations. *Bioinformatics* **32**, 3207-3209 (2016).
11. Marioni, R.E. *et al.* GWAS on family history of Alzheimer's disease. *Translational psychiatry* **8**, 99-99 (2018).
12. Lawlor, D.A. Commentary: Two-sample Mendelian randomization: opportunities and challenges. *International journal of epidemiology* **45**, 908-915 (2016).
13. Bryc, K. *et al.* Genome-wide patterns of population structure and admixture among Hispanic/Latino populations. **107**, 8954-8961 (2010).
14. Conomos, Matthew P. *et al.* Genetic Diversity and Association Studies in US Hispanic/Latino Populations: Applications in the Hispanic Community Health Study/Study of Latinos. *American Journal of Human Genetics* **98**, 165-184 (2016).
15. Disease, G.B.D., Injury, I. & Prevalence, C. Global, regional, and national incidence, prevalence, and years lived with disability for 328 diseases and injuries for 195 countries, 1990-2016: a systematic analysis for the Global Burden of Disease Study 2016. *Lancet (London, England)* **390**, 1211-1259 (2017).
16. Bretz, W.A. *et al.* Longitudinal analysis of heritability for dental caries traits. *Journal of Dental Research* **84**, 1047-1051 (2005).
17. Michalowicz, B.S. *et al.* Evidence of a Substantial Genetic Basis for Risk of Adult Periodontitis. *Journal of Periodontology* **71**, 1699-1707 (2000).
18. Chapple, I.L.C. *et al.* Interaction of lifestyle, behaviour or systemic diseases with dental caries and periodontal diseases: consensus report of group 2 of the joint EFP/ORCA workshop on the boundaries between caries and periodontal diseases. *Journal of Clinical Periodontology* **44**, S39-S51 (2017).
19. Timpson, N.J., Greenwood, C.M.T., Soranzo, N., Lawson, D.J. & Richards, J.B. Genetic architecture: the shape of the genetic contribution to human traits and disease. *Nature Reviews Genetics* **19**, 110 (2017).
20. Nibali, L., Iorio, A.D., Tu, Y.K. & Vieira, A.R. Host genetics role in the pathogenesis of periodontal disease and caries. *Journal of Clinical Periodontology* **44**, S52-S78 (2017).
21. Nibali, L., Di Iorio, A., Tu, Y.-K. & Vieira, A.R. Host genetics role in the pathogenesis of periodontal disease and caries. *Journal of Clinical Periodontology* **44**, S52-S78 (2017).

22. Munz, M. *et al.* A genome-wide association study identifies nucleotide variants at SIGLEC5 and DEFA1A3 as risk loci for periodontitis. *Human Molecular Genetics*, ddy015-ddy015 (2018).
23. Watanabe, K., Taskesen, E., van Bochoven, A. & Posthuma, D. Functional mapping and annotation of genetic associations with FUMA. *Nature Communications* **8**, 1826 (2017).
24. Li, B. *et al.* Evaluation of PrediXcan for prioritizing GWAS associations and predicting gene expression. *Pacific Symposium on Biocomputing. Pacific Symposium on Biocomputing* **23**, 448-459 (2018).
25. Ozawa, Y., Chiba, J. & Sakamoto, S. HLA class II alleles and salivary numbers of mutans streptococci and lactobacilli among young adults in Japan. *Oral Microbiology and Immunology* **16**, 353-357 (2001).
26. Hong, J.H. *et al.* Essential role of carbonic anhydrase XII in secretory gland fluid and HCO₃⁻ secretion revealed by disease causing human mutation. *The Journal of physiology* **593**, 5299-5312 (2015).
27. Esberg, A., Haworth, S., Brunius, C., Lif Holgerson, P. & Johansson, I. Carbonic Anhydrase 6 Gene Variation influences Oral Microbiota Composition and Caries Risk in Swedish adolescents. *Scientific Reports* **9**, 452 (2019).
28. Pers, T.H. *et al.* Biological interpretation of genome-wide association studies using predicted gene functions. *Nature Communications* **6**, 9 (2015).
29. Lek, M. *et al.* Analysis of protein-coding genetic variation in 60,706 humans. *Nature* **536**, 285 (2016).
30. Bowden, J., Davey Smith, G. & Burgess, S. Mendelian randomization with invalid instruments: effect estimation and bias detection through Egger regression. *International Journal of Epidemiology* **44**, 512-525 (2015).
31. O'Connor, L.J. & Price, A.L. Distinguishing genetic correlation from causation across 52 diseases and complex traits. *Nature Genetics* **50**, 1728-1734 (2018).
32. Zhu, Z. *et al.* Causal associations between risk factors and common diseases inferred from GWAS summary data. *Nature Communications* **9**, 224 (2018).

Review Table 1. Impact of excluding HCHS/SOL from LDSR analysis

	Estimated H2 (SE)	Estimated RG with UKB dentures (SE)
DMFS including HCHS/SOL	0.09 (0.018)	0.82 (0.09)
DMFS excluding HCHS/SOL	0.14 (0.04)	0.75 (0.10)

Reviewers' Comments:

Reviewer #1:

Remarks to the Author:

The manuscript is substantially improved in its current format. However, two of my major comments regarding the previous MR-focussed manuscript have been inherited in this new combined manuscript and remain unaddressed, namely:

1) Given the GSMR method is relatively new, it would seem prudent to compare analyses using other available methods (for example, MR-PRESSO, IVW, ...). How robust are these observations when multiple techniques for association testing and heterogeneity assessment are used?

2) Given many of these relationships are bi-directional, how can the authors rule out "common soil" effects where a single trait X is shared between the exposure and outcome? This seems potentially plausible in the context of traits such as cognition and personality. Related to this, can the authors perform multi-variate MR analyses to confirm that all the observed causal associations are independent?

The current disclaimer in the discussion is inappropriate given the expertise of the co-authors in MR and the availability of approaches to explore this in more detail: "We note that the inference drawn from all MR experiments will depend on the approach used to detect and account for horizontal pleiotropy"

Reviewer #3:

Remarks to the Author:

I have reviewed the first paper and am now reviewing this revised version. My main criticism of the first paper related to manuscript style and structure, and I also felt that the two individual papers should be combined into one manuscript. The authors have very well incorporated the suggestions of both reviewers, which resulted in a new version of the manuscript which is better structured, has consistent wording and provides clear information on analyses strategies and results / interpretation. The extensive data that is available with the manuscript will likely inform many downstream studies on function and biology of dental diseases.

I only have two very minor points:

page 10, line 181: if FUMA highlighted several potential genes at this locus, then it would be nice to know some of them (even by direct naming, or linking a Table)

page 10, line 188: the overlap results obtained through the PhenoScanner database are primarily traits for which a large number of GWAS findings (i.e, risk loci) have been identified. I am therefore wondering whether the large number of overlaps between the dental traits and these phenotypes (for instance, height) is just a chance finding reflecting the large number of associated variants for these phenotypes.

Genome-wide analysis of dental caries and periodontitis combining clinical and self-reported data. NCOMMS-18-32981A

Reviewer comments from R1 GWAS paper

Reviewer 1

The manuscript is substantially improved in its current format. However, two of my major comments regarding the previous MR-focussed manuscript have been inherited in this new combined manuscript and remain unaddressed, namely:

1) Given the GSMR method is relatively new, it would seem prudent to compare analyses using other available methods (for example, MR-PRESSO, IVW, ...). How robust are these observations when multiple techniques for association testing and heterogeneity assessment are used?

****Results of sensitivity analyses using 3 different methods were previously located in the supplementary material. We apologise for a lack of overt discussion of these or for clear signposts in the manuscript. We have made efforts to improve this and now include a brief comparison of results from different methods in the main manuscript along with more specific signposts to the relevant section of the supplement containing full results.**

“Sensitivity analysis used alternative estimation tools to explore whether these findings were robust under different modelling assumptions (Methods, Supplementary Note 4). All effect estimates from IVW meta-analysis, MR-Egger regression and a heterogeneity-penalized model-averaging procedure were consistent with the primary estimates within 95% confidence intervals, apart from triglycerides, where the model-averaging procedure suggested a stronger causal effect of DMFS/dentures than indicated by the GSMR primary results (Supplementary Table 11)“ (Results, page 16)

“Estimates from sensitivity analyses were consistent with the estimates from the GSMR primary analysis within 95% confidence intervals for all traits apart from BMI, where the model-averaging procedure supported a stronger causal effect on DMFS/dentures than the GSMR primary estimate (Supplementary Table 13).” (Results, page 16)

We remain confident that the GSMR method performs well as a primary analysis. The ability of this method to produce consistent estimates of causal effect which are not heavily influenced by outlying variants has been demonstrated by applications in both simulated data and complex real-world traits¹.

2) Given many of these relationships are bi-directional, how can the authors rule out “common soil” effects where a single trait X is shared between the exposure and outcome? This seems potentially plausible in the context of traits such as cognition and personality. Related to this, can the authors perform multi-variate MR analyses to confirm that all the observed causal associations are independent?

The current disclaimer in the discussion is inappropriate given the expertise of the co-authors in MR and the availability of approaches to explore this in more detail: “We note that the inference drawn from all MR experiments will depend on the approach used to detect and account for horizontal pleiotropy”

****We agree that dental caries and periodontitis are likely influenced by a range of latent phenotypes and that some of these will be shared with other health traits or outcomes. These latent phenotypes will act as common causes to create both observational and broad genetic correlation between dental diseases and other health traits or outcomes and likely contributed to the large number of genetic correlations reported in the present study.**

We would like to make a distinction between bi-directional relationships and ‘common soil’ effects. We consider bi-directional relationships to be those where trait A affects trait B and trait B affects trait A. These relationships can exist both with and without other common causes or directly shared

underlying biology. In their purest form, the presence of independent instruments of traits A and B allow for a reasonable assertion of the presence, or absence, of bi-directional relationships (for example BMI and C reactive protein²). Conceptually, there is no reason why genetic “instruments” of dental caries and other diseases (which are taken from the lead signals of genetic association for these traits and are ostensibly independent) cannot be used to estimate causal relationships in a bi-directional context using MR methods. This type of approach has been applied in comparable settings, for example physical activity and BMI³ and type 2 diabetes and BMI, used as part of the test bed for GSMR-HEIDI¹.

In practical applications however, there is a natural concern that the instruments used for traits A and B might be highly correlated, therefore changing inference. If a genotype predicts trait A only through its effects on trait C (or indeed as a result of the two phenotypes being poor measures of the same thing), then it must be associated with trait B in the absence of a causal relationship between A and B, as noted by O’Connor⁴. Here, the genotype used to proxy trait ‘A’ fails the exclusion restriction assumption in MR, indicating this is a problem with application of MR and experiment / instrument design. There are conceptually similar problems when a power imbalance between exposure and outcome or poor measurement leads to incorrect inference about the casual direction in a relationship⁵ and where “generously” constructed genomewide predictors are introduced into MR analyses without consideration for their broad predictive properties.

In this case, we believe then that ‘common soil’ effects are likely, but do not prevent valid causal inference or tests for bi-directionality provided that the instruments used for exposure and outcome are valid and non-overlapping. We tried to design the experiment in a way which would be conducive to capturing valid instruments on the one hand and identifying variants which fail the MR assumptions on the other hand. First, the strict cut-off criteria for defining instruments in conjunction with finite statistical power means that the instruments should be enriched for variants with large effect sizes, which are likely to be biologically proximal to dental caries, and this is reflected in the apparent biological function of the loci described in this paper. Next, there is little evidence for enrichment of dental caries loci in any described gene sets, which runs counter to the hypothesis that this set of genotypes captures variation in a latent process which manifests in multiple traits. Finally, invalid instruments will likely have outlying causal effect estimates compared to valid instruments, so should either be detected as heterogeneous variants in GSMR-HEIDI or down-weighted in the heterogeneity-penalized model-averaging procedure.

Despite this, we continue to believe that transparency in methods and results and cautious interpretation are the best protection against invalid inference and have revised the discussion section accordingly. This passage now reads;

“Causal effect estimates in MR experiments may be biased by unaddressed horizontal pleiotropy⁶³ or confounded if both the exposure and outcome are influenced by a latent shared trait with genetic determinants⁶⁴. The different approaches to detect and account for horizontal pleiotropy used in this study yielded similar causal effect estimate; however, lack of precision in these estimates may have masked important differences or evidence for horizontal pleiotropy which will only emerge in future experiments with greater statistical power. Finally, the non-specific nature of the DMFS/dentures phenotype may capture a range of latent traits upstream of clinically manifest disease and tooth loss due to periodontitis, meaning that the downstream effects of DMFS/dentures might involve a wide range of potential mechanisms or mediators.” (Discussion, Page 20)

We assume that the reviewer is referring to multivariable MR, which using summary statistics, has been described to test the direct causal effect for each exposure when multiple exposures with correlated genetic determinants are hypothesized to influence an outcome⁶⁻⁸. These methods are not suitable for simultaneous estimation of the causal effect of a single exposure on multiple outcomes, where a network approach⁹ would typically be used to explore questions of mediation or an MR pheWAS analysis may be more fitting. We have reservations in applying this type of approach here, partly for reasons of outcome overlap (as discussed immediately above) and given the limited variance in exposure explained. In the reciprocal direction, we have used the multivariable MR IVW estimator⁷ to estimate the direct casual effects of BMI and fasting glucose (the two traits which

showed some evidence of effect through causal estimates) on dental caries. These analyses produced similar effect estimates to the univariate IVW estimates (Review Table 1). These results have been added to the supplementary information, with a brief interpretation and flag in the manuscript.

“Additional sensitivity analysis using multivariable MR suggested that BMI and fasting blood glucose had independent causal effects on dental disease (Supplementary Table 14)” (Results, Page 16).

With respect to the disclaimer statement pointed out by the reviewer, we take the point that it is important to be direct about the implications of findings from the analysis undertaken. However, whilst we are experienced in the approaches used, we are not keen to overemphasise any one result nor do we wish to claim that any particular analytical approach is optimal. We have attempted to present a full analysis in a manner which is transparent and a discussion which addresses the main results of this work in a conservative manner. We do hope that the additions made have improved this part of the paper.

Reviewer 3

I have reviewed the first paper and am now reviewing this revised version. My main criticism of the first paper related to manuscript style and structure, and I also felt that the two individual papers should be combined into one manuscript. The authors have very well incorporated the suggestions of both reviewers, which resulted in a new version of the manuscript which is better structured, has consistent wording and provides clear information on analyses strategies and results / interpretation. The extensive data that is available with the manuscript will likely inform many downstream studies on function and biology of dental diseases.

I only have two very minor points:

page 10, line 181: if FUMA highlighted several potential genes at this locus, then it would be nice to know some of them (even by direct naming, or linking a Table)

**There were 25 genes, which we felt was too large a number to present as a meaningful list. Instead, we have added a new supplementary table as suggested which gives more information about the possible candidate genes at this locus (Supplementary Table 4).

page 10, line 188: the overlap results obtained through the PhenoScanner database are primarily traits for which a large number of GWAS findings (i.e, risk loci) have been identified. I am therefore wondering whether the large number of overlaps between the dental traits and these phenotypes (for instance, height) is just a chance finding reflecting the large number of associated variants for these phenotypes.

** Results obtained through PhenoScanner will be biased towards phenotypes with many reported associations. Both the sample size and number of investigations is higher for traits like height than other complex traits. We have revised the discussion item to make this clearer;

“Cross-trait comparisons of dental diseases with other traits at single-variant level are only possible when GWAS results are available for any given trait and well-powered investigations are likely to report more associations than small studies. Together, this means that the list of observed overlaps between dental diseases and other traits will be biased towards well-investigated traits and outcomes.” Discussion, pages 19-20.

Despite this limitation, we believe the overlap analysis is a useful addition to the manuscript and have continued to include these results. Specifically for height, we believe this finding is plausible given that height is associated with both periodontitis and caries in observational studies^{10,11}. Here it is possible to speculate that there are common influences on skeletal and dental development during active growth (such as protein availability) or that inflammatory events which present as infections and impaired growth in childhood present as periodontitis later in life. As the reviewer mentioned

height only to illustrate a general point about PhenoScanner and as these interpretations are speculative, we have not added any discussion about height to the manuscript.

Review Table 1

Exposure	Outcome	Univariable IVW beta (se)	MV IVW beta (se)	95% confidence intervals for MV IVW beta	MV IVW P
BMI	DMFS/dentures	0.14 (0.010)	0.13 (0.010)	0.108, 0.149	4.5×10^{-38}
Fasting glucose		0.047 (0.023)	0.061 (0.022)	0.017, 0.105	0.006

References

1. Zhu, Z. *et al.* Causal associations between risk factors and common diseases inferred from GWAS summary data. *Nat. Commun.* **9**, 224 (2018).
2. Timpson, N.J. *et al.* C-reactive protein levels and body mass index: elucidating direction of causation through reciprocal Mendelian randomization. *Int. J. Obesity (2005)* **35**, 300-308 (2011).
3. Richmond, R.C. *et al.* Assessing Causality in the Association between Child Adiposity and Physical Activity Levels: A Mendelian Randomization Analysis. *PLoS Med.* **11**, e1001618 (2014).
4. O'Connor, L.J. & Price, A.L. Distinguishing genetic correlation from causation across 52 diseases and complex traits. *Nat. Genet.* **50**, 1728-1734 (2018).
5. Hemani, G., Tilling, K. & Davey Smith, G. Orienting the causal relationship between imprecisely measured traits using GWAS summary data. *PLoS Genet.* **13**, e1007081 (2017).
6. Sanderson, E., Davey Smith, G., Windmeijer, F. & Bowden, J. An examination of multivariable Mendelian randomization in the single-sample and two-sample summary data settings. *Int. J. Epidemiol.* (2018).
7. Burgess, S. & Thompson, S.G. Multivariable Mendelian randomization: the use of pleiotropic genetic variants to estimate causal effects. *Am. J. Epidemiol.* **181**, 251-260 (2015).
8. Rees, J.M.B., Wood, A.M. & Burgess, S. Extending the MR-Egger method for multivariable Mendelian randomization to correct for both measured and unmeasured pleiotropy. *Stat. Med.* **36**, 4705-4718 (2017).
9. Burgess, S., Daniel, R.M., Butterworth, A.S., Thompson, S.G. & Consortium, E.P.-I. Network Mendelian randomization: using genetic variants as instrumental variables to investigate mediation in causal pathways. *Int. J. Epidemiol.* **44**, 484-495 (2015).
10. Chakravathy, P.K., Chenna, D. & Chenna, V.J.E.A.o.P.D. Association of anthropometric measures and dental caries among a group of adolescent cadets of Udipi district, South India. *Eur. Arch. Paed. Dent.* **13**, 256-260 (2012).
11. Meisel, P., Kohlmann, T. & Kocher, T. Association of height with inflammation and periodontitis: the Study of Health in Pomerania. *J. Clin. Periodontol.* **34**, 390-396 (2007).

Reviewers' Comments:

Reviewer #1:

Remarks to the Author:

I think the changes made to the discussion are appropriate and I have no further concerns.